# Hierarchical clustering with dot products recovers hidden tree structure

**Annie Gray**
School of Mathematics
University of Bristol, UK
annie.gray@bristol.ac.uk

**Alexander Modell**
Department of Mathematics
Imperial College London, UK
a.modell@imperial.ac.uk

**Patrick Rubin-Delanchy**
School of Mathematics
University of Bristol, UK
patrick.rubin-delanchy@bristol.ac.uk

**Nick Whiteley**
School of Mathematics
University of Bristol, UK
nick.whiteley@bristol.ac.uk

## Abstract

In this paper we offer a new perspective on the well established agglomerative clustering algorithm, focusing on recovery of hierarchical structure. We recommend a simple variant of the standard algorithm, in which clusters are merged by maximum average dot product and not, for example, by minimum distance or within-cluster variance. We demonstrate that the tree output by this algorithm provides a bona fide estimate of generative hierarchical structure in data, under a generic probabilistic graphical model. The key technical innovations are to understand how hierarchical information in this model translates into tree geometry which can be recovered from data, and to characterise the benefits of simultaneously growing sample size and data dimension. We demonstrate superior tree recovery performance with real data over existing approaches such as UPGMA, Ward's method, and HDBSCAN.

## 1 Introduction

Hierarchical structure is known to occur in many natural and man-made systems [32], and the problem considered in this paper is how to recover this structure from data. Hierarchical clustering algorithms, [25, 36, 37, 41] are very popular techniques which organise data into nested clusters, used routinely by data scientists and machine learning researchers, and are easily accessible through open source software packages such as scikit-learn [38]. We focus on perhaps the most popular family of such techniques: agglomerative clustering [17], [24, Ch.3], in which clusters of data points are merged recursively.

Agglomerative clustering methods are not model-based procedures, but rather simple algorithms. Nevertheless, in this work we uncover a new perspective on agglomerative clustering by introducing a general form of generative statistical model for the data, but without assuming specific parametric families of distributions (e.g., Gaussian). In our model (section 2.1), hierarchy takes the form of a tree defining the conditional independence structure of latent variables, using elementary concepts from probabilistic graphical modelling [30, 28]. In a key innovation, we then augment this conditional independence tree to form what we call a *dendrogram*, whose geometry is related to population statistics of the data. The new insight which enables tree recovery in our setting (made precise and explained in section 3) is that *dot products between data vectors reveal heights of most recent common ancestors in the dendrogram*.

We suggest an agglomerative algorithm which merges clusters according to highest sample average dot product (section 2.2). This is in contrast to many existing approaches which quantify dissimilarity

37th Conference on Neural Information Processing Systems (NeurIPS 2023).

between data vectors using Euclidean distance. We also consider the case where data are preprocessed by reducing dimension using PCA. We mathematically analyse the performance of our dot product clustering algorithm and establish that under our model, with sample size $n$ and data dimension $p$ growing simultaneously at appropriate rates, the merge distortion [20, 27] between the algorithm output and underlying tree vanishes. In numerical examples with real data (section 4), where we have access to labels providing a notion of true hierarchy, we compare performance of our algorithm against existing methods in terms of a Kendall $\tau_b$ correlation performance measure, which quantifies association between ground-truth and estimated tree structure. We examine statistical performance with and without dimension reduction by PCA, and illustrate how dot products versus Euclidean distances relate to semantic structure in ground-truth hierarchy.

**Related work** Agglomerative clustering methods combine some dissimilarity measure with a 'linkage' function, determining which clusters are combined. Popular special cases include UPGMA [44] and Ward's method [48], against which we make numerical comparisons (section 4). Owing to the simple observation that, in general, finding the largest dot product between data vectors is not equivalent to finding the smallest Euclidean distance, we can explain why these existing methods may not correctly recover tree structure under our model (appendix E). Popular density-based clustering methods methods include CURE [22], OPTICS [7], BIRCH [52] and HDBSCAN [9]. In section 4 we discuss the extent to which these methods can and cannot be compared against ours in terms of tree recovery performance.

Existing theoretical treatments of hierarchical clustering involve different mathematical problem formulations and assumptions to ours. One common setup is to assume an underlying ultrametric space whose geometry specifies the unknown tree, and/or to study tree recovery as $n \to \infty$ with respect to a cost function, e.g, [10, 16, 43, 11, 14, 15, 31, 12]. An alternative problem formulation addresses recovery of the cluster tree of the probability density from which it is assumed data are sampled [42, 20, 27]. The unknown tree in our problem formulation specifies conditional independence structure, and so it has a very different interpretation to the trees in all these cited works. Moreover, our data are vectors in $\mathbb{R}^p$, and $p \to \infty$ is a crucial aspect of our convergence arguments, but in the above cited works $p$ plays no role or is fixed. Our definition of dendrogram is different to that in e.g. [10]: we do not require all leaf vertices to be equidistant from the root – a condition which arises as the "fixed molecular clock" hypothesis [18, 34] in phylogenetics. We also allow data to be sampled from non-leaf vertices. There is an enormous body of work on tree reconstruction methods in phylogenetics, e.g. listed at [4], but these are mostly not general-purpose solutions to the problem of inferring hierarchy. Associated theoretical convergence results are limited in scope, e.g, the famous work [18] is limited to a fixed molecular clock, five taxa and does not allow observation error.

## 2 Model and algorithm

### 2.1 Statistical model, tree and dendrogram

Where possible, we use conventional terminology from the field of probabilistic graphical models, e.g., [28] to define our objects and concepts. Our model is built around an unobserved tree $\mathcal{T} = (\mathcal{V}, \mathcal{E})$, that is a directed acyclic graph with vertex and edge sets $\mathcal{V}$ and $\mathcal{E}$, with two properties: $\mathcal{T}$ is connected (ignoring directions of edges), and each vertex has at most one parent, where we say $u$ is a parent of $v$ if there is an edge from $u$ to $v$. We observe data vectors $\mathbf{Y}_i \in \mathbb{R}^p$, $i = 1, \dots, n$, which we model as:

$$\mathbf{Y}_i = \mathbf{X}(Z_i) + \mathbf{S}(Z_i)\mathbf{E}_i, \tag{1}$$

comprising three independent sources of randomness:

- $Z_1, \dots, Z_n$ are i.i.d., discrete random variables, with distribution supported on a subset of vertices $\mathcal{Z} \subseteq \mathcal{V}$, $|\mathcal{Z}| < \infty$;
- $\mathbf{X}(v) := [X_1(v) \; \cdots \; X_p(v)]^\top$ is an $\mathbb{R}^p$-valued random vector for each vertex $v \in \mathcal{V}$;
- $\mathbf{E}_1, \dots, \mathbf{E}_n$ are i.i.d, $\mathbb{R}^p$-valued random vectors. The elements of $\mathbf{E}_i$ are i.i.d., zero mean and unit variance. For each $z \in \mathcal{Z}$, $\mathbf{S}(z) \in \mathbb{R}^{p \times p}$, is a deterministic matrix.

For each $v \in \mathcal{Z}$, one can think of the vector $\mathbf{X}(v)$ as the random centre of a "cluster", with correlation structure of the cluster determined by the matrix $\mathbf{S}(z)$. The latent variable $Z_i$ indicates which cluster

the $i$th data vector $\mathbf{Y}_i$ is associated with. The vectors $\mathbf{X}(v)$, $v \in \mathcal{V} \setminus \mathcal{Z}$, correspond to unobserved vertices in the underlying tree. As an example, $\mathcal{Z}$ could be the set of leaf vertices of $\mathcal{T}$, but neither our methods nor theory require this to be the case. Throughout this paper, we assume that the tree $\mathcal{T}$ determines two distributional properties of $\mathbf{X}$. Firstly, we assume $\mathcal{T}$ is the conditional independence graph of the collection of random variables $X_j := \{X_j(v); v \in \mathcal{V}\}$ for each $j$, that is,

**A1.** *for all $j = 1, \ldots, p$, the marginal probability density or mass function of $X_j$ factorises as:*

$$p(x_j) = \prod_{v \in \mathcal{V}} p\left(x_j(v) | x_j(\mathrm{Pa}_v)\right),$$

*where $\mathrm{Pa}_v$ denotes the parent of vertex $v$.*

However we do not necessarily require that $X_1, \ldots, X_p$ are independent or identically distributed. Secondly, we assume

**A2.** *for each $j = 1 \ldots, p$, the following martingale-like property holds:*

$$\mathbb{E}\left[X_j(v) | X_j(\mathrm{Pa}_v)\right] = X_j(\mathrm{Pa}_v),$$

*for all vertices $v \in \mathcal{V}$ except the root.*

The conditions **A1** and **A2** induce an additive hierarchical structure in $\mathbf{X}$. For any distinct vertices $u, v \in \mathcal{V}$ and $w$ an ancestor of both, **A1** and **A2** imply that given $X_j(w)$, the increments $X_j(u) - X_j(w)$ and $X_j(v) - X_j(w)$ are conditionally independent and both conditionally mean zero.

To explain our algorithm we need to introduce the definition of a *dendrogram*, $\mathcal{D} = (\mathcal{T}, h)$, where $h : \mathcal{V} \to \mathbb{R}_+$ is a function which assigns a height to each vertex of $\mathcal{T}$, such that $h(v) \geq h(\mathrm{Pa}_v)$ for any vertex $v \in \mathcal{V}$ other than the root. The term "dendrogram" is derived from the ancient Greek for "tree" and "drawing", and indeed the numerical values $h(v)$, $v \in \mathcal{V}$, can be used to construct a drawing of $\mathcal{T}$ where height is measured with respect to some arbitrary baseline on the page, an example is shown in figure 1(a). With $\langle \cdot, \cdot \rangle$ denoting the usual dot product between vectors, the function

$$\alpha(u, v) := \frac{1}{p} \mathbb{E}\left[\langle \mathbf{X}(u), \mathbf{X}(v) \rangle\right], \quad u, v \in \mathcal{V}, \tag{2}$$

will act as a measure of affinity underlying our algorithm. The specific height function we consider is $h(v) := \alpha(v, v)$. The martingale property **A2** ensures that this height function satisfies $h(v) \geq h(\mathrm{Pa}_v)$ as required, see lemma 2 in appendix C. In appendix E we also consider cosine similarity as an affinity measure, and analyse its performance under a multiplicative noise model, cf. the additive model (1).

## 2.2 Algorithm

Combining the model (1) with (2) we have $\alpha(Z_i, Z_j) = p^{-1}\mathbb{E}[\langle \mathbf{Y}_i, \mathbf{Y}_j \rangle | Z_i, Z_j]$ for $i \neq j \in [n]$, $[n] := \{1, \ldots, n\}$. The input to algorithm 1 is an estimate $\hat{\alpha}(\cdot, \cdot)$ of all the pairwise affinities $\alpha(Z_i, Z_j)$, $i \neq j \in [n]$. We consider two approaches to estimating $\alpha(Z_i, Z_j)$, with and without dimension reduction by uncentered PCA. For some $r \leq \min\{p, n\}$, let $\mathbf{V} \in \mathbb{R}^{p \times r}$ denote the matrix whose columns are orthonormal eigenvectors of $\sum_{i=1}^n \mathbf{Y}_i \mathbf{Y}_i^\top$ associated with its $r$ largest eigenvalues. Then $\zeta_i := \mathbf{V}^\top \mathbf{Y}_i$ is $r$-dimensional vector of principal component scores for the $i$th data vector. The two possibilities for $\hat{\alpha}(\cdot, \cdot)$ we consider are:

$$\hat{\alpha}_{\mathrm{data}}(i, j) := \frac{1}{p} \langle \mathbf{Y}_i, \mathbf{Y}_j \rangle, \qquad \hat{\alpha}_{\mathrm{pca}}(i, j) := \frac{1}{p} \langle \zeta_i, \zeta_j \rangle. \tag{3}$$

In the case of $\hat{\alpha}_{\mathrm{pca}}$ the dimension $r$ must be chosen. Our theory in section 3.2 assumes that $r$ is chosen as the rank of the matrix with entries $\alpha(u, v)$, $u, v \in \mathcal{Z}$, which is at most $|\mathcal{Z}|$. In practice $r$ usually must be chosen based on the data, we discuss this in appendix A.

Algorithm 1 returns a dendrogram $\hat{\mathcal{D}} = (\hat{\mathcal{T}}, \hat{h})$, comprising a tree $\hat{\mathcal{T}} = (\hat{\mathcal{V}}, \hat{\mathcal{E}})$ and height function $\hat{h}$. Each vertex in $\hat{\mathcal{V}}$ is a subset of $[n]$, thus indexing a subset of the data vectors $\mathbf{Y}_1, \ldots, \mathbf{Y}_n$. The leaf vertices are the singleton sets $\{i\}$, $i \in [n]$, corresponding to the data vectors themselves. As algorithm 1 proceeds, vertices are appended to $\hat{\mathcal{V}}$, edges are appended to $\hat{\mathcal{E}}$, and the domain of the function $\hat{\alpha}(\cdot, \cdot)$ is extended as affinities between elements of $\hat{\mathcal{V}}$ are computed. Throughout the paper we simplify notation by writing $\hat{\alpha}(i, j)$ as shorthand for $\hat{\alpha}(\{i\}, \{j\})$ for $i, j \in [n]$, noting that each argument of $\hat{\alpha}(\cdot, \cdot)$ is in fact a subset of $[n]$.

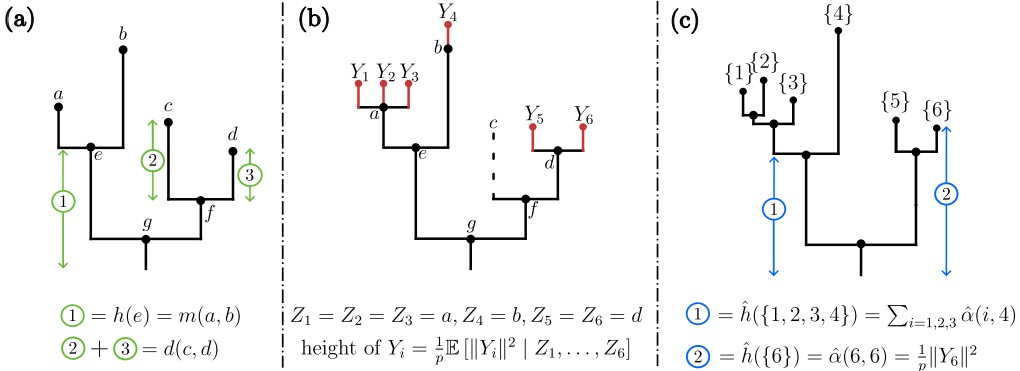

Figure 1: (a) An example of the dendrogram $\mathcal{D}$ with $\mathcal{V} = \{a, b, c, d, e, f, g\}$, see lemma 1 for interpretation of the merge height $m(\cdot, \cdot)$ and distance $d(\cdot, \cdot)$. Horizontal distances in this diagram are chosen arbitrarily. (b) $\mathcal{D}$ augmented according to $\mathcal{Z} = \{a, b, c, d\}$ and the realization: $Z_1, Z_2, Z_3 = a$, $Z_4 = b$, $Z_5 = Z_6 = d$, see section 3.3 for discussion. (c) The dendrogram $\hat{\mathcal{D}}$ output from algorithm 1 in the case $\hat{\alpha} = \hat{\alpha}_{\text{data}}$. $\hat{\mathcal{D}}$ can be seen to approximate the dendrogram in (b) and hence $\mathcal{D}$.

---

**Algorithm 1** Dot product hierarchical clustering

**Input:** pairwise affinities $\hat{\alpha}(\cdot, \cdot)$ between $n$ data points

1: Initialise partition $P_0 := \{\{1\}, \ldots, \{n\}\}$, vertex set $\hat{\mathcal{V}} := P_0$ and edge set $\hat{\mathcal{E}}$ to the empty set.
2: **for** $m \in \{1, \ldots, n-1\}$ **do**
3:      Find distinct pair $u, v \in P_{m-1}$ with largest affinity
4:      Update $P_{m-1}$ to $P_m$ by merging $u, v$ to form $w := u \cup v$
5:      Append vertex $w$ to $\hat{\mathcal{V}}$ and directed edges $w \to u$ and $w \to v$ to $\hat{\mathcal{E}}$
6:      Define affinity between $w$ and other members of $P_m$ as

$$\hat{\alpha}(w, \cdot) := \frac{|u|}{|w|}\hat{\alpha}(u, \cdot) + \frac{|v|}{|w|}\hat{\alpha}(v, \cdot),$$

     and $\hat{\alpha}(w, w) := \hat{\alpha}(u, v)$.
7: **end for**

**Output:** Dendrogram $\hat{\mathcal{D}} := (\hat{\mathcal{T}}, \hat{h})$, comprising tree $\hat{\mathcal{T}} = (\hat{\mathcal{V}}, \hat{\mathcal{E}})$ and heights $\hat{h}(v) := \hat{\alpha}(v, v)$ for $v \in \hat{\mathcal{V}} \setminus P_0$, and $\hat{h}(v) := \max\{\hat{h}(\text{Pa}_v), \hat{\alpha}(v, v)\}$ for $v \in P_0$.

---

**Implementation using scikit-learn** The presentation of algorithm 1 has been chosen to simplify its theoretical analysis, but alternative formulations of the same method may be much more computationally efficient in practice. In appendix B we outline how algorithm 1 can easily be implemented using the `AgglomerativeClustering` class in scikit-learn [38].

## 3 Performance Analysis

### 3.1 Merge distortion is upper bounded by affinity estimation error

In order to explain the performance of algorithm 1 we introduce the *merge height* functions:

$$m(u, v) := h(\text{most recent common ancestor of } u \text{ and } v), \qquad u, v \in \mathcal{V},$$
$$\hat{m}(u, v) := \hat{h}(\text{most recent common ancestor of } u \text{ and } v), \qquad u, v \in \hat{\mathcal{V}}.$$

To simplify notation we write $\hat{m}(i, j)$ as shorthand for $\hat{m}(\{i\}, \{j\})$, for $i, j \in [n]$. The discrepancy between any two dendrograms whose vertices are in correspondence can be quantified by *merge distortion* [20] – the maximum absolute difference in merge height across all corresponding pairs of

vertices. [20] advocated merge distortion as a performance measure for cluster-tree recovery, which is different to our model-based formulation, but merge distortion turns out to be a useful and tractable performance measure in our setting too. As a preface to our main theoretical results, lemma 1 explains how the geometry of the dendrogram $\mathcal{D}$, in terms of merge heights, is related to population statistics of our model. Defining $d(u,v) := h(u) - h(w) + h(v) - h(w)$ for $u \neq v \in \mathcal{V}$, where $w$ is the most recent common ancestor of $u$ and $v$, we see $d(u,v)$ is the vertical distance on the dendrogram from $u$ down to $w$ then back up to $v$, as illustrated in figure 1(a).

**Lemma 1.** *For any two vertices $u, v \in \mathcal{V}$,*

$$m(u,v) = \frac{1}{p}\mathbb{E}\left[\langle \mathbf{X}(u), \mathbf{X}(v)\rangle\right] = \alpha(u,v), \quad d(u,v) = \frac{1}{p}\mathbb{E}\left[\|\mathbf{X}(u) - \mathbf{X}(v)\|^2\right]. \quad (4)$$

The proof is in appendix C. Considering the first two equalities in (4), it is natural to ask if the estimated affinities $\hat{\alpha}(\cdot,\cdot)$ being close to the true affinities $\alpha(\cdot,\cdot)$ implies a small merge distortion between $\hat{m}(\cdot,\cdot)$ and $m(\cdot,\cdot)$. This is the subject of our first main result, theorem 1 below. In appendix E we use the third equality in (4) to explain why popular agglomerative techniques such as UPGMA [44] and Ward's method [48] which merge clusters based on proximity in Euclidean distance may enjoy limited success under our model, but in general do not correctly recover tree structure.

Let $b$ denote the minimum branch length of $\mathcal{D}$, that is, $b = \min\{h(v) - h(\text{Pa}_v)\}$, where the minimum is taken over all vertices in $\mathcal{V}$ except the root.

**Theorem 1.** *Let the function $\hat{\alpha}(\cdot,\cdot)$ given as input to algorithm 1 be real-valued and symmetric but otherwise arbitrary. For any $z_1, \ldots, z_n \in \mathcal{Z}$, if*

$$\max_{i,j\in[n],i\neq j} |\alpha(z_i, z_j) - \hat{\alpha}(i,j)| < b/2,$$

*then the dendrogram returned by algorithm 1 satisfies*

$$\max_{i,j\in[n],i\neq j} |m(z_i, z_j) - \hat{m}(i,j)| \leq \max_{i,j\in[n],i\neq j} |\alpha(z_i, z_j) - \hat{\alpha}(i,j)|. \quad (5)$$

The proof is in appendix C.

## 3.2 Affinity estimation error vanishes with increasing dimension and sample size

Our second main result, theorem 2 below, concerns the accuracy of estimating the affinities $\alpha(\cdot,\cdot)$ using $\hat{\alpha}_{\text{data}}$ or $\hat{\alpha}_{\text{pca}}$ as defined in (3). We shall consider the following technical assumptions.

**A3** (Mixing across dimensions). *For mixing coefficients $\varphi$ satisfying $\sum_{k\geq 1} \varphi^{1/2}(k) < \infty$ and all $u, v \in \mathcal{Z}$, the sequence $\{(X_j(u), X_j(v)); j \geq 1\}$ is $\varphi$-mixing.*

**A4** (Bounded moments). *For some $q \geq 2$, $\sup_{j\geq 1} \max_{v\in\mathcal{Z}} \mathbb{E}[|X_j(v)|^{2q}] < \infty$ and $\mathbb{E}[|\mathbf{E}_{11}|^{2q}] < \infty$, where $\mathbf{E}_{11}$ is the first element of the vector $\mathbf{E}_1$.*

**A5** (Disturbance control). $\max_{v\in\mathcal{Z}} \|\mathbf{S}(v)\|_{\text{op}} \in O(1)$ *as $p \to \infty$, where $\|\cdot\|_{\text{op}}$ is the spectral norm.*

**A6** (PCA rank). *The dimension $r$ chosen in definition of $\hat{\alpha}_{\text{pca}}$, see (3), is equal to the rank of the matrix with entries $\alpha(u,v)$, $u, v \in \mathcal{Z}$.*

The concept of $\varphi$-mixing is a classical weak-dependence condition, e.g. [19, 39]. **A3** implies that for each $j \geq 1$, $(X_j(u), X_j(v))$ and $(X_{j+\delta}(u), X_{j+\delta}(v))$ are asymptotically independent as $\delta \to \infty$. However, it is important to note that $\hat{\alpha}_{\text{data}}$ and $\hat{\alpha}_{\text{pca}}$ in (3), and hence the operation of algorithm 1 with these inputs, are invariant to permutation of the data dimensions $j = 1, \ldots, p$. Thus our analysis under **A3** only requires there is *some* permutation of dimensions under which $\varphi$-mixing holds. **A4** is a fairly mild integrability condition. **A5** allows control of magnitudes of the "disturbance" vectors $\mathbf{Y}_i - \mathbf{X}(Z_i) = \mathbf{S}(Z_i)\mathbf{E}_i$. Further background and discussion of assumptions is given in appendix C.2.

**Theorem 2.** *Assume that A1-A5 hold and let $q$ be as in A4. Then*

$$\max_{i,j\in[n],i\neq j} |\alpha(Z_i, Z_j) - \hat{\alpha}_{\text{data}}(i,j)| \in O_{\mathbb{P}}\left(\frac{n^{2/q}}{\sqrt{p}}\right). \quad (6)$$

*If additionally A3 is strengthened from $\varphi$-mixing to independence, $\mathbf{S}(v) = \sigma\mathbf{I}_p$ for some constant $\sigma \geq 0$ and all $v \in \mathcal{Z}$ (in which case A5 holds), and A6 holds, then*

$$\max_{i,j\in[n],i\neq j} |\alpha(Z_i, Z_j) - \hat{\alpha}_{\text{pca}}(i,j)| \in O_{\mathbb{P}}\left(\sqrt{\frac{nr}{p}} + \sqrt{\frac{r}{n}}\right). \quad (7)$$

The proof of theorem 2 is in appendix C.2. We give an original and self-contained proof of (6). To prove (7) we use a recent uniform-concentration result for principal component scores from [49]. Overall, theorem 2 says that affinity estimation error vanishes if the dimension $p$ grows faster enough relative to $n$ (and $r$ in the case of $\hat{\alpha}_{\text{pca}}$, noting that under **A6**, $r \leq |\mathcal{Z}|$, so it is sensible to assume $r$ is much smaller than $n$ and $p$). The argument of $O_{\mathbb{P}}(\cdot)$ is the convergence rate; if $(X_{p,n})$ is some collection of random variables indexed by $p$ and $n$, $X_{p,n} \in O_{\mathbb{P}}(n^{2/q}/\sqrt{p})$ means that for any $\epsilon > 0$, there exists $\delta$ and $M$ such that $n^{2/q}/\sqrt{p} \geq M$ implies $\mathbb{P}(|X_{p,n}| > \delta) < \epsilon$. We note the convergence rate in (6) is decreasing in $q$ where as rate in (7) is not. It is an open mathematical question whether (7) can be improved in this regard; sharpening the results of Whiteley et al. [49] used in the proof of (7) seems very challenging. However, when $q = 2$, (6) gives $O_{\mathbb{P}}(n/\sqrt{p})$ compared to $O_{\mathbb{P}}\left(\sqrt{nr/p} + \sqrt{r/n}\right)$ in (7), i.e. an improvement from $n$ to $\sqrt{nr}$ in the first term. We explore empirical performance of $\hat{\alpha}_{\text{data}}$ versus $\hat{\alpha}_{\text{pca}}$ in section 4.2.

## 3.3 Interpretation

By combining theorems 1 and 2, we find that when $b > 0$ is constant and $\hat{\alpha}$ is either $\hat{\alpha}_{\text{data}}$ or $\hat{\alpha}_{\text{pca}}$, the merge distortion

$$\max_{i,j \in [n], i \neq j} |m(Z_i, Z_j) - \hat{m}(i,j)| \tag{8}$$

converges to zero at rates given by the r.h.s of (6) and (7). To gain intuition into what (8) tells us about the resemblance between $\hat{\mathcal{D}}$ and $\mathcal{D}$, it is useful to consider an intermediate dendrogram illustrated in figure 1(b) which conveys the realized values of $Z_1, \ldots, Z_n$. This dendrogram is constructed from $\mathcal{D}$ by adding a leaf vertex corresponding to each observation $\mathbf{Y}_i$, with parent $Z_i$ and height $p^{-1}\mathbb{E}[\|\mathbf{Y}_i\|^2 | Z_1, \ldots, Z_n] = h(Z_i) + p^{-1}\text{tr}[\mathbf{S}(Z_i)^\top \mathbf{S}(Z_i)]$, and deleting any $v \in \mathcal{Z}$ such that $Z_i \neq v$ for all $i \in [n]$ (e.g., vertex $c$ in figure 1(b)). The resulting merge height between the vertices corresponding to $\mathbf{Y}_i$ and $\mathbf{Y}_j$ is $m(Z_i, Z_j)$. (8) being small implies this must be close to $\hat{m}(i,j)$ in $\hat{\mathcal{D}}$ as in figure 1(c). Moreover, in the case $\hat{\alpha} = \hat{\alpha}_{\text{data}}$, the height $\hat{h}(\{i\})$ of leaf vertex $\{i\}$ in $\hat{\mathcal{D}}$ is upper bounded by $p^{-1}\|\mathbf{Y}_i\|^2$, which under our statistical assumptions is concentrated around $p^{-1}\mathbb{E}[\|\mathbf{Y}_i\|^2 | Z_1, \ldots, Z_n]$, i.e., the heights of the leaves in figure 1(c) approximate those of the corresponding leaves in figure 1(b).

Overall we see that $\hat{\mathcal{D}}$ in figure 1(c) approximates the dendrogram in figure 1(b), and in turn $\mathcal{D}$. However even if $m(Z_i, Z_j) = \hat{m}(i,j)$ for all $i, j$, the tree output from algorithm 1, $\hat{\mathcal{T}}$, may not be isomorphic (i.e., equivalent up to relabelling of vertices) to the tree in figure 1(b); $\hat{\mathcal{T}}$ is always binary and has $2n - 1$ vertices, whereas the tree in figure 1(b) may not be binary, depending on the underlying $\mathcal{T}$ and the realization of $Z_1, \ldots, Z_n$. This reflects the fact that merge distortion, in general, is a *pseudometric* on dendrograms. However, if one restricts attention to specific classes of true dendrograms $\mathcal{D}$, for instance binary trees with non-zero branch lengths, then asymptotically algorithm 1 can recover them exactly. We explain this point further in appendix C.3.

## 4 Numerical experiments

We explore the numerical performance of algorithm 1 in the setting of five data sets summarised below. The real datasets used are open source, and full details of data preparation and sources are given in appendix D.

**Simulated data.** A simple tree structure with vertices $\mathcal{V} = \{1, 2, 3, 4, 5, 6, 7, 8\}$, edge set $\mathcal{E} = \{6 \rightarrow 1, 6 \rightarrow 2, 6 \rightarrow 3, 7 \rightarrow 4, 7 \rightarrow 5, 8 \rightarrow 6, 8 \rightarrow 7\}$ and $\mathcal{Z} = \{1, 2, 3, 4, 5\}$ (the leaf vertices). $Z_1, \ldots, Z_n$ are drawn from the uniform distribution on $\mathcal{Z}$. The $X_j(v)$ are Gaussian random variables, independent across $j$. Full details of how these variables are sampled are in appendix D. The elements of $\mathbf{E}_i$ are standard Gaussian, and $\mathbf{S}(v) = \sigma \mathbf{I}_p$ with $\sigma = 1$.

**20 Newsgroups.** We used a random subsample of $n = 5000$ documents from the well-known 20 Newsgroups data set [29]. Each data vector corresponds to one document, capturing its $p = 12818$ Term Frequency Inverse Document Frequency features. The value of $n$ was chosen to put us in the regime $p \geq n$, to which our theory is relevant – see section 3.2. Some ground-truth labelling of documents is known: each document is associated with 1 of 20 newsgroup topics, organized at two hierarchical levels.

**Zebrafish gene counts.** These data comprise gene counts in zebrafish embryo cells taken from their first day of development [47]. As embryos develop, cells differentiate into various types with specialised, distinct functions, so the data are expected to exhibit tree-like structure mapping these changes. We used a subsample such that $n = 5079$ and $p = 5498$ to put us in the $p \geq n$ regime. Each cell has two labels: the tissue that the cell is from and a subcategory of this.

**Amazon reviews.** This dataset contains customer reviews on Amazon products [1]. A random sample of $n = 5000$ is taken and each data vector corresponds to one review with $p = 5594$ Term Frequency Inverse Document Frequency features. Each product reviewed has labels which make up a three-level hierarchy of product types.

**S&P 500 stock returns.** The data are $p = 1259$ daily returns between for $n = 368$ stocks which were constituents of the S&P 500 market index [2] between 2013 to 2018. The two-level hierarchy of stock sectors by industries and sub-industries follows the Global Industry Classification Standard [3].

### 4.1 Comparing algorithm 1 to existing methods

We numerically compare algorithm 1 against three very popular variants of agglomerative clustering: UPGMA with Euclidean distance, Ward's method, and UPGMA with cosine distance. These are natural comparators because they work by iteratively merging clusters in a manner similar to algorithm 1, but using different criteria for choosing which clusters to merge. In appendix E we complement our numerical results with mathematical insights into how these methods perform under our modelling assumptions. Numerical results for other linkage functions and distances are given in appendix D. Several popular density-based clustering methods use some hierarchical structure, such as CURE [22], OPTICS [7] and BIRCH [52] but these have limitations which prevent direct comparisons: they aren't equipped with a way to simplify the structure into a tree, which it is our aim to recover, and only suggest extracting a flat partition based on a density threshold. HDBSCAN [9] is a density-based method that doesn't have these limitations, and we report numerical comparisons against it.

**Kendall $\tau_b$ ranking correlation.** For real data some ground-truth hierarchical labelling may be available but ground-truth merge heights usually are not. We need a performance measure to quantitatively compare methods operating on such data. Commonly used clustering performance measures such as the Rand index [40] and others [23, 21] allow pairwise comparisons between partitions, but do not capture information about hierarchical structure. The cophenetic correlation coefficient [45] is commonly used to compare dendrograms, but relies on an assumption that points close in Euclidean distance should be considered similar which is incompatible with our notion of dot product affinity. To overcome these obstacles we formulate a performance measure as follows. For each of $n$ data points, we rank the other $n - 1$ data points according to the order in which they merge with it in the ground-truth hierarchy. We then compare these ground truth rankings to those obtained from a given hierarchical clustering algorithm using the Kendall $\tau_b$ correlation coefficient [26]. This outputs a value in the interval $[-1, 1]$, with $-1, 1$ and $0$ corresponding to negative, positive and lack of association between the ground-truth and algorithm-derived rankings. We report the mean association value across all $n$ data points as the overall performance measure. Table 1 shows results with raw data vectors $\mathbf{Y}_{1:n}$ or PC scores $\zeta_{1:n}$ taken as input to the various algorithms. For all the data sets except S&P 500, algorithm 1 is found to recover hierarchy more accurately than other methods. We include the results for the S&P 500 data to give a balanced scientific view, and in appendix E we discuss why our modelling assumptions may not be appropriate for these data, thus explaining the limitations of algorithm 1.

### 4.2 Simulation study of dot product estimation with and without PCA dimension reduction

For high-dimensional data, reducing dimension with PCA prior to clustering may reduce overall computational cost. Assuming $\zeta_{1:n}$ are obtained from, e.g., a partial SVD, in time $O(npr)$, the time complexity of evaluating $\hat{\alpha}_{\text{pca}}$ is $O(npr + n^2r)$, versus $O(n^2p)$ for $\hat{\alpha}_{\text{data}}$, although this ignores the cost of choosing $r$. In table 1 we see for algorithm 1, the results for input $\mathbf{Y}_{1:n}$ are very similar to those for $\zeta_{1:n}$. To examine this more closely and connect our findings to theorem 2, we now compare $\hat{\alpha}_{\text{data}}$ and $\hat{\alpha}_{\text{pca}}$ as estimates of $\alpha$ through simulation. The model is as described at the start of section 4. In figure 2(a)-(b), we see that when $p$ is growing with $n$, and when $p$ is constant, the $\hat{\alpha}_{\text{pca}}$ error is very slightly smaller than the $\hat{\alpha}_{\text{data}}$ error. By contrast, in figure 2(c), when $n = 10$ is

Table 1: Kendall $\tau_b$ ranking performance measure. For the dot product method, i.e., algorithm 1, $\mathbf{Y}_{1:n}$ as input corresponds to using $\hat{\alpha}_{\text{data}}$, and $\zeta_{1:n}$ corresponds to $\hat{\alpha}_{\text{pca}}$. The mean Kendall $\tau_b$ correlation coefficient is reported alongside the standard error (numerical value shown is the standard error $\times 10^3$).

| Data | Input | Dot product | UPGMA w/ cos. dist. | HDBSCAN | UPGMA w/ Eucl. dist. | Ward |
|---|---|---|---|---|---|---|
| Newsgroups | $\mathbf{Y}_{1:n}$ | 0.26 (2.9) | 0.26 (2.9) | -0.010 (0.65) | 0.23 (2.7) | 0.18 (2.5) |
| | $\zeta_{1:n}$ | 0.24 (2.6) | 0.18 (1.9) | -0.016 (1.9) | 0.038 (1.5) | 0.19 (2.7) |
| Zebrafish | $\mathbf{Y}_{1:n}$ | 0.34 (3.4) | 0.25 (3.1) | 0.023 (2.9) | 0.27 (3.2) | 0.30 (3.8) |
| | $\zeta_{1:n}$ | 0.34 (3.4) | 0.27 (3.2) | 0.11 (2.8) | 0.16 (2.5) | 0.29 (3.8) |
| Reviews | $\mathbf{Y}_{1:n}$ | 0.15 (2.5) | 0.12 (1.9) | 0.014 (1.1) | 0.070 (1.5) | 0.10 (1.8) |
| | $\zeta_{1:n}$ | 0.14 (2.4) | 0.14 (2.4) | -0.0085 (0.78) | 0.14 (2.6) | 0.12 (2.4) |
| S&P 500 | $\mathbf{Y}_{1:n}$ | 0.34 (10) | 0.34 (10) | 0.14 (9.3) | 0.34 (1) | 0.35 (10) |
| | $\zeta_{1:n}$ | 0.36 (9.4) | 0.42 (11) | 0.33 (13) | 0.39 (11) | 0.39 (11) |
| Simulated | $\mathbf{Y}_{1:n}$ | 0.86 (1) | 0.81 (2) | 0.52 (8) | 0.52 (8) | 0.52 (8) |
| | $\zeta_{1:n}$ | 0.86 (1) | 0.81 (2) | 0.52 (8) | 0.52 (8) | 0.52 (8) |

fixed, we see that the $\hat{\alpha}_{\text{pca}}$ error is larger than that for $\hat{\alpha}_{\text{data}}$. This inferior performance of $\hat{\alpha}_{\text{pca}}$ for very small and fixed $n$ is explained by $n$ appearing in the denominator of the second term in the rate $O_{\mathbb{P}}(\sqrt{nr/p}+\sqrt{r/n})$ for $\hat{\alpha}_{\text{pca}}$ in theorem 2 versus $n$ appearing only in the numerator of $O_{\mathbb{P}}(n^{2/q}/\sqrt{p})$ for $\hat{\alpha}_{\text{data}}$. Since it is Gaussian, this simulation model has finite exponential-of-quadratic moments, which is a much stronger condition than **A4**; we conjecture the convergence rate in this Gaussian case is $O_{\mathbb{P}}(\sqrt{\log n/p})$ for $\hat{\alpha}_{\text{data}}$, which would be consistent with figure 2(a). These numerical results seem to suggest the rate for $\hat{\alpha}_{\text{pca}}$ is similar, thus the second result of theorem 2 may not be sharp.

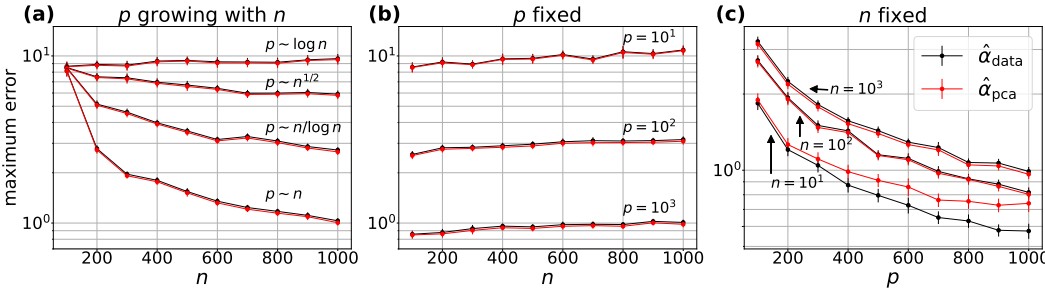

Figure 2: Simulation study of $\hat{\alpha}_{\text{data}}$ and $\hat{\alpha}_{\text{pca}}$ as estimators of $\alpha$. All three subplots display the maximum error, $\max_{i,j \in [n], i \neq j} |\alpha(Z_i, Z_j) - \hat{\alpha}(i,j)|$, for $\hat{\alpha} = \hat{\alpha}_{\text{data}}$ (black in all subplots (a)-(c)) and $\hat{\alpha} = \hat{\alpha}_{\text{pca}}$ (red). Error bars showing the standard deviation from 100 simulations are present for all data points, but in some cases are so small they are barely visible.

### 4.3 Comparing dot product affinities and Euclidean distances for the 20 Newsgroups data

In this section we expand on the results in table 1 for the 20 Newsgroups data, by exploring how inter-topic and intra-topic dot product affinities and Euclidean distances relate to ground-truth labels. Most existing agglomerative clustering techniques quantify dissimilarity using Euclidean distance. To compare dot products and Euclidean distances, figures 3(a)-(b) show, for each topic, the top five topics with the largest average dot product and smallest average Euclidean distance respectively. We see that clustering of semantically similar topic classes is apparent when using dot products but not when using Euclidean distance. Note that the average dot product affinity between comp.windows.x and itself is not shown in figure 3(b), but is shown in 3(a), by the highest dark blue square in the comp.windows.x column. In appendix D we provide additional numerical results illustrating that with $n$ fixed, performance in terms of $\tau_b$ correlation coefficient increases with $p$.

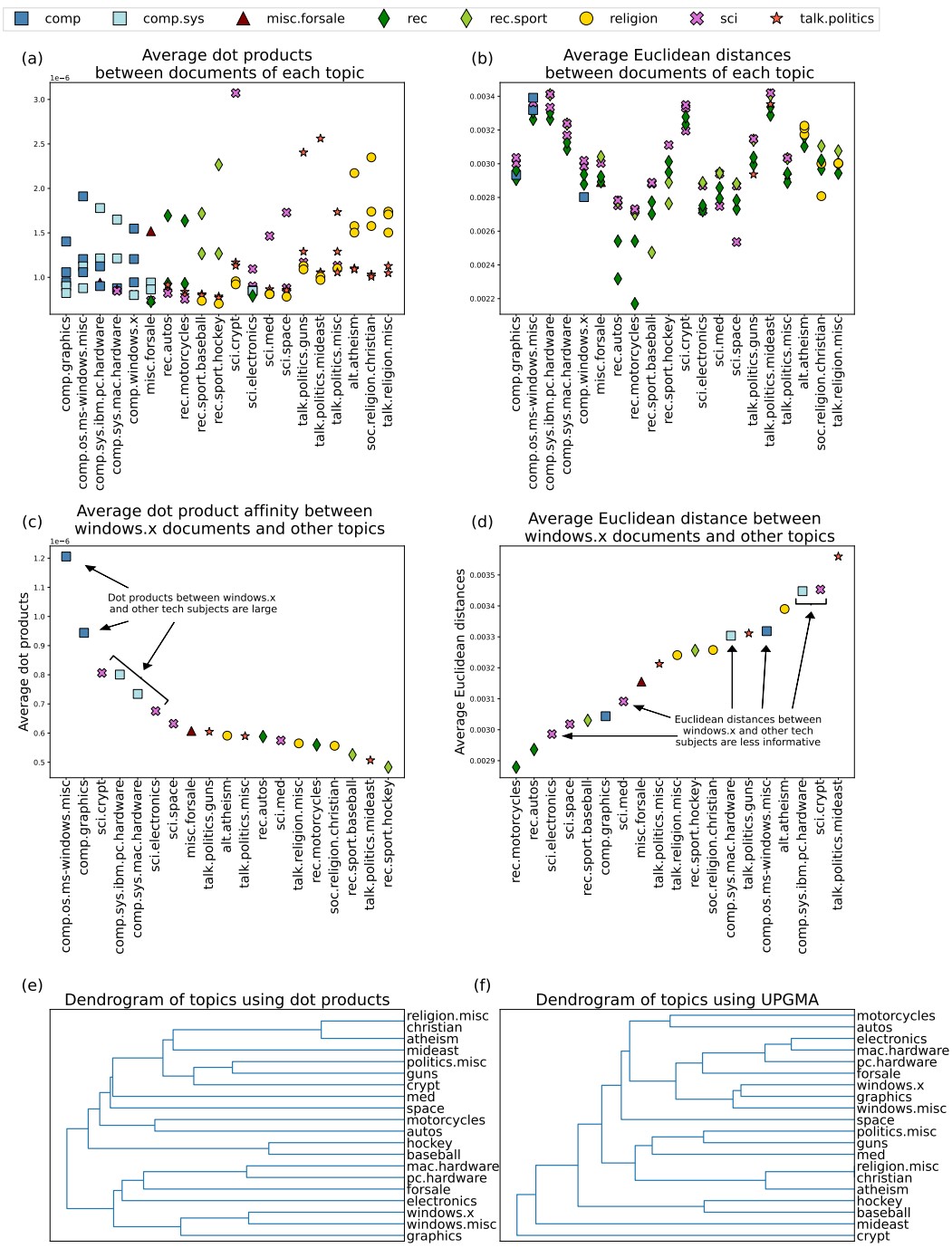

Figure 3: Analysis of the 20 Newsgroups data. Marker shapes correspond to newsgroup classes and marker colours correspond to topics within classes. The first/second columns show results for dot products/Euclidean distances respectively. First row: for each topic ($x$-axis), the affinity/distance ($y$-axis) to the top five best-matching topics, calculated using average linkage of PC scores between documents within topics. Second row: average affinity/distance between documents labelled 'comp.windows.x' and all other topics. Third row: dendrograms output from algorithm 1 and UPGMA applied to cluster topics.

For one topic ('comp.windows.x') the results are expanded in figures 3(c)-(d) to show the average dot products and average Euclidean distances to all other topics. Four out of the five topics with the largest dot product affinity belong to the same 'comp' topic class and other one is a semantically similar 'sci.crypt' topic. Whereas, the other topics in the same 'comp' class are considered dissimilar in terms of Euclidean distance.

In order to display visually compact estimated dendrograms, we applied algorithm 1 and UPGMA in a semi-supervised setting where each topic is assigned its own PC score, taken to be the average of the PC scores of the documents in that topic, and then the algorithms are applied to cluster the topics. The results are shown in figures 3(e)-(f) (for ease of presentation, leaf vertex 'heights' are fixed to be equal).

## 5 Limitations and opportunities

Our algorithm is motivated by modelling assumptions. If these assumptions are not appropriate for the data at hand, then the algorithm cannot be expected to perform well. A notable limitation of our model is that $\alpha(u, v) \geq 0$ for all $u, v \in \mathcal{V}$ (see lemma 3 in appendix C). This is an inappropriate assumption when there are strong negative cross-correlations between some pairs of data vectors, and may explain why our algorithm has inferior performance on the S&P 500 data in table 1. Further discussion is given in appendix E. A criticism of agglomerative clustering algorithms in their basic form is that their computational cost scales faster than $O(n^2)$. Approximations to standard agglomerative methods which improve computational scalability have been proposed [33, 5, 35]. Future research could investigate analogous approximations and speed-up of our method. Fairness in hierarchical clustering has been recently studied in cost function-based settings by [6] and in greedy algorithm settings by [13]. Future work could investigate versions of our algorithm which incorporate fairness measures.

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

# Appendices

## A Wasserstein dimension selection

In order to compute the principal component scores $\zeta_1, \ldots, \zeta_n$, one must choose the dimension $r$. Traditionally and somewhat heuristically, this is done by finding the "elbow" in the scree plot of eigenvalues. The bias-variance tradeoff associated with choosing $r$ was explored by [49], who suggested a data-splitting method of dimension selection for high-dimensional data using Wasserstein distances. Whilst this method may be more costly than the traditional "elbow" approach, it was empirically demonstrated in [49] to have superior performance.

---

**Algorithm 2** Wasserstein PCA dimension selection [49]

---

**Input:** data vectors $\mathbf{Y}_1, \ldots, \mathbf{Y}_n \in \mathbb{R}^p$.

1: **for** $r \in \{1, \ldots, \min(n, p)\}$ **do**
2:     Let $\mathbf{V} \in \mathbb{R}^{p \times r}$ denote the matrix whose columns are orthonormal eigenvectors associated with the $r$ largest eigenvalues of $\sum_{i=1}^{\lceil n/2 \rceil} \mathbf{Y}_i \mathbf{Y}_i^\top$
3:     Orthogonally project $\mathbf{Y}_1, \ldots, \mathbf{Y}_{\lceil n/2 \rceil}$ onto the column space of $\mathbf{V}$, $\hat{\mathbf{Y}}_i \coloneqq \mathbf{V}\mathbf{V}^\top \mathbf{Y}_i$
4:     Compute Wasserstein distance $d_r$ between $\hat{\mathbf{Y}}_i, \ldots, \hat{\mathbf{Y}}_{\lceil n/2 \rceil}$ and $\mathbf{Y}_{\lceil n/2 \rceil + 1}, \ldots, \mathbf{Y}_n$ (as point sets in $\mathbb{R}^p$)
5: **end for**

**Output:** selected dimension $\hat{r} = \operatorname{argmin} \{d_r\}$.

---

We note that in practice, the eigenvectors appearing in this procedure could be computed sequentially as $r$ grows, and to limit computational cost one might consider $r$ only up to some $r_{\max} < \min(n, p)$.

## B Implementation using scikit-learn

Algorithm 1 can be implemented using the Python module scikit-learn [38] via their `AgglomerativeClustering` class, using the standard 'average' linkage criterion and a custom metric function to compute the desired affinities. However, `AgglomerativeClustering` merges clusters based on minimum distance metric, whereas algorithm 1 merges according to maximum dot product. Therefore, the custom metric function we used in our implementation calculates all the pairwise dot products and subtracts them from the maximum. This transformation needs to then be rectified if accessing the merge heights.

All code released as part of this paper is under the MIT License and can be found at `https://github.com/anniegray52/dot_product_hierarchical`

## C Proofs and supporting theoretical results

**Lemma 2.** *The height function $h(v) \coloneqq \alpha(v, v)$ satisfies $h(v) \geq h(\mathrm{Pa}_v)$ for all $v \in \mathcal{V}$ except the root.*

*Proof.*

$$h(v) = \frac{1}{p} \mathbb{E}[\|\mathbf{X}(v) - \mathbf{X}(\mathrm{Pa}_v) + \mathbf{X}(\mathrm{Pa}_v)\|^2]$$

$$= \frac{1}{p} \mathbb{E}[\|\mathbf{X}(v) - \mathbf{X}(\mathrm{Pa}_v)\|^2] + 2\frac{1}{p} \mathbb{E}[\langle \mathbf{X}(v) - \mathbf{X}(\mathrm{Pa}_v), \mathbf{X}(\mathrm{Pa}_v) \rangle] + h(\mathrm{Pa}_v) \geq h(\mathrm{Pa}_v),$$

where **A2** combined with the tower property and linearity of conditional expectation implies $\mathbb{E}[\langle \mathbf{X}(v) - \mathbf{X}(\mathrm{Pa}_v), \mathbf{X}(\mathrm{Pa}_v) \rangle] = 0$. $\qquad \square$

**Lemma 3.** *For all $u, v \in \mathcal{V}$, $\alpha(u, v) \geq 0$.*

*Proof.* If $u = v$, $\alpha(u, v) \geq 0$ holds immediately from the definition of $\alpha$ in (2). For $u \neq v$ with most recent common ancestor $w$,

$$
\mathbb{E}[\langle \mathbf{X}(u), \mathbf{X}(v) \rangle] = \sum_{j=1}^{p} \mathbb{E}\left[\mathbb{E}[X_j(u)X_j(v)|X_j(w)]\right]
$$
$$
= \sum_{j=1}^{p} \mathbb{E}\left[\mathbb{E}[X_j(u)|X_j(w)]\mathbb{E}[X_j(v)|X_j(w)]\right]
$$
$$
= \sum_{j=1}^{p} \mathbb{E}[|X_j(w)|^2] \geq 0,
$$

where the second equality uses **A1** together with standard conditional independence aguments, and the third equality uses **A2**.

$\square$

*Proof of lemma 1.* Let $w$ be the most recent common ancestor of $u$ and $v$. For each $j = 1, \ldots, p$, the property **A1** together with standard conditional independence arguments imply that $X_j(u)$ and $X_j(v)$ are conditionally independent given $X_j(w)$, and the property **A2** implies that $\mathbb{E}[X_j(u)|X_j(w)] = \mathbb{E}[X_j(v)|X_j(w)] = X_j(w)$. Therefore, by the tower property of conditional expectation,

$$
\mathbb{E}[X_j(u)X_j(v)] = \mathbb{E}\left[\mathbb{E}[X_j(u)X_j(v)|X_j(w)]\right]
$$
$$
= \mathbb{E}\left[\mathbb{E}[X_j(u)|X_j(w)]\mathbb{E}[X_j(v)|X_j(w)]\right]
$$
$$
= \mathbb{E}[X_j(w)^2].
$$

Hence, using the definitions of the merge height $m$, the height $h$ and the affinity $\alpha$,

$$
m(u, v) = h(w) = \alpha(w, w) = \frac{1}{p} \sum_{j=1}^{p} \mathbb{E}[X_j(w)^2] = \frac{1}{p} \sum_{j=1}^{p} \mathbb{E}[X_j(u)X_j(v)] = \alpha(u, v),
$$

which proves the first equality in the statement. The second equality is the definition of $\alpha$.

For the third equality in the statement, we have

$$
d(u, v) = h(u) + h(v) - 2h(w)
$$
$$
= \alpha(u, u) + \alpha(v, v) - 2\alpha(u, v)
$$
$$
= \frac{1}{p}\mathbb{E}\left[\langle \mathbf{X}(u), \mathbf{X}(u) \rangle\right] + \frac{1}{p}\mathbb{E}\left[\langle \mathbf{X}(v), \mathbf{X}(v) \rangle\right] - 2\frac{1}{p}\mathbb{E}\left[\langle \mathbf{X}(u), \mathbf{X}(v) \rangle\right]
$$
$$
= \frac{1}{p}\mathbb{E}\left[\|\mathbf{X}(u) - \mathbf{X}(v)\|^2\right],
$$

where the first equality uses the definition of $d$, and the second equality uses the definition of $h$ and $h(w) = m(u, v) = \alpha(u, v)$.

$\square$

## C.1 Proof of Theorem 1

The following lemma establishes an identity concerning the affinities computed in algorithm 1 which will be used in the proof of theorem 1.

**Lemma 4.** *Let $P_m$, $m \geq 0$, be the sequence of partitions of $[n]$ constructed in algorithm 1 . Then for any $m \geq 0$,*

$$
\hat{\alpha}(u, v) = \frac{1}{|u||v|} \sum_{i \in u, j \in v} \hat{\alpha}(i, j), \quad \text{for all distinct pairs } u, v \in P_m. \tag{9}
$$

*Proof.* The proof is by induction on $m$. With $m = 0$, (9) holds immediately since $P_0 = \{\{1\}, \ldots, \{n\}\}$. Now suppose (9) holds at step $m$. Then for any distinct pair $w, w' \in P_{m+1}$,

either $w$ or $w'$ is the result of merging two elements of $P_m$, or $w$ and $w'$ are both elements of $P_m$. In the latter case the induction hypothesis immediately implies:

$$\hat{\alpha}(w, w') = \frac{1}{|w||w'|} \sum_{i \in w, j \in w'} \hat{\alpha}(i, j).$$

In the case that $w$ or $w'$ is the result of a merge, suppose w.l.o.g. that $w = u \cup v$ for some $u, v \in P_m$ and $w' \in P_m$. Then by definition of $\hat{\alpha}$ in algorithm 1 ,

$$\begin{aligned}
\hat{\alpha}(w, w') &= \frac{|u|}{|w|} \hat{\alpha}(u, w') + \frac{|v|}{|w|} \hat{\alpha}(v, w') \\
&= \frac{|u|}{|w|} \frac{1}{|u||w'|} \sum_{i \in u, j \in w'} \hat{\alpha}(i, j) + \frac{|v|}{|w|} \frac{1}{|v||w'|} \sum_{i \in v, j \in w'} \hat{\alpha}(i, j) \\
&= \frac{1}{|w||w'|} \sum_{i \in w, j \in w'} \hat{\alpha}(i, j),
\end{aligned}$$

where the final equality uses $w = u \cup v$. The induction hypothesis thus holds at step $m + 1$. $\qquad \square$

The following proposition establishes the validity of the height function constructed in algorithm 1. Some of the arguments used in this proof are qualitatively similar to those used to study reducible linkage functions by, e.g., Sumengen et al. [46], see also historical references therein.

**Proposition 1.** *With $\hat{\mathcal{V}}$ the vertex set and $\hat{h}$ the height function constructed in algorithm 1 with any symmetric, real-valued input $\hat{\alpha}(\cdot, \cdot)$, it holds that $\hat{h}(v) \geq \hat{h}(\mathrm{Pa}_v)$ for all vertices $v \in \hat{\mathcal{V}}$ except the root.*

*Proof.* The required inequality $\hat{h}(v) \geq \hat{h}(\mathrm{Pa}_v)$ holds immediately for all the leaf vertices $v \in P_0 = \{\{1\}, \ldots, \{n\}\}$ by the definition of $\hat{h}$ in algorithm 1. All the remaining vertices in the output tree, i.e., those in $\hat{\mathcal{V}} \setminus P_0$, are formed by merges over the course of the algorithm. For $m \geq 0$ let $w_m = u_m \cup v_m$ denote the vertex formed by merging some $u_m, v_m \in P_m$. Then $w_m = \mathrm{Pa}_{u_m}$ and $w_m = \mathrm{Pa}_{v_m}$. Each $u_m$ is either a member of $P_0$ or equal to $w_{m'}$ for some $m' < m$. The same is true of each $v_m$. It therefore suffices to show that $\hat{h}(w_m) \geq \hat{h}(w_{m+1})$ for $m \geq 0$, where by definition in the algorithm, $\hat{h}(w_m) = \hat{\alpha}(u_m, v_m)$. Also by definition in the algorithm, $\hat{h}(w_{m+1})$ is the largest pairwise affinity between elements of $P_{m+1}$. Our objective therefore is to upper-bound this largest affinity and compare it to $\hat{h}(w_m) = \hat{\alpha}(u_m, v_m)$.

The affinity between $w_m = u_m \cup v_m$ and any other element $w'$ of $P_{m+1}$ (which must also be an element of $P_m$) is, by definition in the algorithm,

$$\begin{aligned}
\hat{\alpha}(w_m, w') &= \frac{|u_m|}{|u_m| + |v_m|} \hat{\alpha}(u_m, w') + \frac{|v_m|}{|u_m| + |v_m|} \hat{\alpha}(v_m, w') \\
&\leq \max\{\hat{\alpha}(u_m, w'), \hat{\alpha}(v_m, w')\} \\
&\leq \hat{\alpha}(u_m, v_m),
\end{aligned}$$

where the last inequality holds because $u_m, v_m$, by definition, have the largest affinity amongst all elements of $P_m$. For the same reason, the affinity between any two distinct elements of $P_{m+1}$ neither of which is $w_m$ (and therefore both of which are elements of $P_m$) is upper-bounded by $\hat{\alpha}(u_m, v_m)$. We have therefore established $\hat{h}(w_m) = \hat{\alpha}(u_m, v_m) \geq \hat{h}(w_{m+1})$ as required, and this completes the proof.

$\qquad \square$

*Proof of theorem 1.* Let us introduce some definitions used throughout the proof.

$$M := \max_{i \neq j} |\hat{\alpha}(i, j) - \alpha(z_i, z_j)|. \tag{10}$$

We arbitrarily chose and then fix $i, j \in [n]$ with $i \neq j$, and define

$$H := m(z_i, z_j), \qquad \hat{H} := \hat{m}(i, j). \tag{11}$$

Let $u$ denote the most recent common ancestor of the leaf vertices $\{i\}$ and $\{j\}$ in $\hat{\mathcal{D}}$ and let $m \geq 1$ denote the step of the algorithm at which $u$ is created by a merge, that is $m = \min\{m' \geq 1 : u \in P_{m'}\}$. We note that by construction, $u$ is equal to the union of all leaf vertices with ancestor $u$, and by definition of $\hat{h}$ in algorithm 1 $\hat{h}(u) = \hat{H}$.

Let $v$ denote the most recent common ancestor of $z_i$ and $z_j$ in $\mathcal{D}$, which has height $h(v) = H$.

**Lower bound on $m(z_i, z_j) - \hat{m}(i, j)$.** There is no partition of $u$ into two non-empty sets $A, B \subseteq [n]$ such that $\hat{\alpha}(k, l) < \hat{H}$ for all $k \in A$ and $l \in B$. We prove this by contradiction. Suppose that such a partition exists. There must be a step $m' \leq m$ at which some $A' \subseteq A$ is merged some $B' \subseteq B$. The vertex $w$ formed by this merge would have height

$$
\begin{aligned}
\hat{h}(w) &= \hat{\alpha}(A', B') \\
&= \frac{1}{|A'||B'|} \sum_{k \in A', l \in B'} \hat{\alpha}(k, l) < \hat{H} = \hat{h}(u),
\end{aligned}
$$

where the first equality is the definition of $\hat{h}(w)$ in the algorithm and the second equality holds by lemma 4. However, in this construction $u$ is an ancestor of $w$, and $\hat{h}(w) < \hat{h}(u)$ therefore contradicts the result of proposition 1.

As a device to be used in the next step of the proof, consider an undirected graph with vertex set $u$, in which there is an edge between two vertices $k$ and $l$ if and only if $\hat{\alpha}(k, l) \geq \hat{H}$. Then, because there is no partition as established above, this graph must be connected. Now consider a second undirected graph, also with vertex set $u$, in which there is any edge between two vertices $k$ and $l$ if and only if $\alpha(z_k, z_l) \geq \hat{H} - M$. Due to the definition of $M$ in (10), any edge in the first graph is an edge in the second, so the second graph is connected too. Let $k$, $l$, and $\ell$ be any distinct members of $u$. Using the fact established in lemma 1 that $\alpha(z_k, z_l)$ and $\alpha(z_l, z_\ell)$ are respectively the merge heights in $\mathcal{D}$ between $z_k$ and $z_l$, and $z_l$ and $z_\ell$, it can be seen that if there are edges between $k$ and $l$ and between $l$ and $\ell$ in the second graph, there must also be an edge in that graph between $k$ and $\ell$. Combined with the connectedness, this implies that the second graph is complete, so that $\alpha(z_k, z_l) \geq \hat{H} - M$ for all distinct $k, l \in u$. In particular $\alpha(z_i, z_j) \geq \hat{H} - M$, and since $m(z_i, z_j) = \alpha(z_i, z_j)$, we find

$$
m(z_i, z_j) - \hat{m}(i, j) \geq -M. \tag{12}
$$

**Upper bound on $m(z_i, z_j) - \hat{m}(i, j)$.** Let $S_v = \{i \in [n] : z_i = v \text{ or } z_i \text{ has ancestor } v \text{ in } \mathcal{D}\}$. For $k, l \in S_v$, lemma 1 tells us $\alpha(z_k, z_l)$ is the merge height between $z_k$ and $z_l$, so $\alpha(z_k, z_l) \geq H$. Using (10), we therefore have

$$
\hat{\alpha}(k, l) \geq H - M, \quad \forall k, l \in S_v. \tag{13}
$$

It follows from the definition of $\hat{h}$ in the algorithm that if $S_v = [n]$, the heights of all vertices in $\hat{\mathcal{D}}$ are greater than or equal to $H - M$. This implies $\hat{H} \geq H - M$. In summary, we have shown that when $S_v = [n]$,

$$
m(z_i, z_j) - \hat{m}(i, j) \leq M. \tag{14}
$$

It remains to consider the case $S_v \neq [n]$. The proof of the same upper bound (14) in this case is more involved. In summary, we need to establish that the most recent common ancestor of $\{i\}$ and $\{j\}$ in $\hat{\mathcal{D}}$ has height at least $H - M$. The main idea of the proof is to consider the latest step of the algorithm at which a vertex with height at least $H - M$ is formed by a merge, and show the partition formed by this merge contains the most recent common ancestor of $\{i\}$ and $\{j\}$, or an ancestor thereof.

To this end let $m^*$ denote the latest step in algorithm 1 at which the vertex formed, $w^*$, has height greater than or equal to $H - M$. To see that $m^*$ must exist, notice

$$
\max_{k \neq l \in [n]} \hat{\alpha}(k, l) \geq \alpha(z_i, z_j) - M, \tag{15}
$$

by definition of $M$ in (10). Combined with the definition of $\hat{h}$ in algorithm 1, the vertex formed by the merge at step 1 of the algorithm therefore has height greater than or equal to $H - M$. Therefore $m^*$ is indeed well-defined.

Our next objective is to show that the partition $P_{m^*}$ formed at step $m^\star$ contains an element which itself contains both $i$ and $j$. We proceed by establishing some facts about $S_v$ and $P_{m^*}$.

Let $\bar{S}_v := [n] \setminus S_v$. For $k \in S_v$, $l \in \bar{S}_v$, $v$ cannot be an ancestor of $z_l$, by lemma 1 $\alpha(z_k, z_l)$ is the merge height of $z_k$ and $z_l$, and $b$ is the minimum branch length in $\mathcal{D}$, so we have $\alpha(z_k, z_l) \leq H - b$. From (10) we then find

$$\hat{\alpha}(k, l) \leq H - b + M, \quad \forall\, k \in S_v, l \in \bar{S}_v. \tag{16}$$

We claim that no element of $P_{m^*}$ can contain both an element of $S_v$ and an element of $\bar{S}_v$. We prove this claim by contradiction. If such an element of $P_{m^*}$ did exist, there would be a step $m' \leq m^*$ at which some $A' \subseteq S_v$ is merged with some $B' \subseteq \bar{S}_v$. But the vertex $w'$ formed by this merge would be assigned height $\hat{h}(w') = \hat{\alpha}(A', B') \leq H - b + M < H - M$, where the first inequality uses lemma 4 and (16), and the second inequality uses the assumption of the theorem that $M < b/2$. Recalling the definition of $w^*$ we have $\hat{h}(w^*) \geq H - M$. We therefore see that $w^*$ is an ancestor of $w'$ with $\hat{h}(w^*) > \hat{h}(w')$, contradicting the result of proposition 1.

Consider the elements of $P_{m^*}$, denoted $A$ and $B$, which contain $i$ and $j$ respectively. We claim that $A = B$. We prove this claim by contradiction. Suppose $A \neq B$. As established in the previous paragraph, neither $A$ nor $B$ can contain an element of $\bar{S}_v$. Therefore, using lemma 4 and (13),

$$\hat{\alpha}(A, B) = \frac{1}{|A||B|} \sum_{k \in A, l \in B} \hat{\alpha}(k, l) \geq H - M.$$

Again using the established fact that no element of $P_{m^*}$ can contain both an element of $S_v$ and an element of $\bar{S}_v$, $m^*$ cannot be the final step of the algorithm, since that would require $P_{m^*} = \{[n]\}$. Therefore $\hat{\alpha}(A, B)$ is one of the affinities which algorithm 1 would maximise over at step $m^* + 1$, so the height of the vertex formed by a merge at step $m^* + 1$ would be greater than or equal to $H - M$, which contradicts the definition of $m^*$. Thus we have proved there exists an element of $P_{m^*}$ which contains both $i$ and $j$. This element must be the most recent common ancestor of $\{i\}$ and $\{j\}$, or an ancestor thereof. Also, this element must have been formed by a merge at a step less than or equal to $m^*$ and so must have height greater than or equal to $H - M$. Invoking proposition 1 we have thus established $\hat{H} \geq H - M$. In summary, in the case $S_v \neq [n]$, we have shown

$$m(z_i, z_j) - \hat{m}(i, j) \leq M. \tag{17}$$

Combining the lower bound (12) with the upper bounds (14), (17) and the fact that $i, j$ were chosen arbitrarily, completes the proof.

$\square$

## C.2   Supporting material and proof for Theorem 2

**Definitions and interpretation for assumptions A3 and A5**

We recall the definition of $\varphi$-mixing from, e.g., [39]. For a sequence of random variables $\{\xi_j; j \geq 1\}$, define:

$$\varphi(k) := \sup_{j \geq 1} \sup_{A \in \mathcal{F}_1^j, B \in \mathcal{F}_{j+k}^\infty, \mathbb{P}(A) > 0} |\mathbb{P}(B|A) - \mathbb{P}(B)|.$$

where $\mathcal{F}_i^j$ is the $\sigma$-algebra generated by $\xi_i, \ldots, \xi_j$. Then $\{\xi_j; j \geq 1\}$ is said to be $\varphi$-mixing if $\varphi(k) \searrow 0$ as $k \to \infty$.

To interpret assumption **A5** notice

$$\mathbb{E}[\|\mathbf{S}(Z_i)\mathbf{E}_i\|^2 | Z_1, \ldots, Z_n] \leq \|\mathbf{S}(Z_i)\|_{\mathrm{op}}^2 \mathbb{E}[\|\mathbf{E}_i\|^2] \leq \max_{v \in \mathcal{Z}} \|\mathbf{S}(v)\|_{\mathrm{op}}^2 \, p \, \mathbb{E}[|\mathbf{E}_{11}|^2],$$

where the first inequality uses the independence of $\mathbf{E}_i$ and $Z_i$ and the second inequality uses the fact that the elements of the vectors $\mathbf{E}_i$ are i.i.d. Since $\mathbf{Y}_i - \mathbf{X}(Z_i) = \mathbf{S}(Z_i)\mathbf{E}_i$, **A5** thus implies $\mathbb{E}[\|\mathbf{Y}_i - \mathbf{X}(Z_i)\|^2] \in O(p)$ as $p \to \infty$, which can be viewed as a natural growth rate since $p$ is the dimension of the disturbance vector $\mathbf{Y}_i - \mathbf{X}(Z_i)$. In the proof of proposition 2 below, **A5** is used in a similar manner to control dot products of the form $\langle \mathbf{Y}_i - \mathbf{X}_i, \mathbf{X}_j \rangle$ and $\langle \mathbf{Y}_i - \mathbf{X}_i, \mathbf{Y}_j - \mathbf{X}_j \rangle$.

*Proof of Theorem 2.* For the first claim of the theorem, proposition 2 combined with the tower property of conditional expectation imply that for any $\delta > 0$,

$$\mathbb{P}\left(\max_{1 \le i < j \le n} \left|p^{-1} \langle \mathbf{Y}_i, \mathbf{Y}_j \rangle - \alpha(Z_i, Z_j)\right| \ge \delta\right)$$

$$\le \frac{1}{\delta^q} \frac{1}{p^{q/2}} \frac{n(n-1)}{2} C(q, \varphi) M(q, \mathbf{X}, \mathbf{E}, \mathbf{S}), \quad (18)$$

from which (6) follows.

The second claim of the theorem is in essence a corollary to [49][Thm 1]. A little work is needed to map the setting of the present work on to the setting of Whiteley et al. [49][Thm 1]. To see the connection, we endow the finite set $\mathcal{Z}$ in the present work with the discrete metric: $d_{\mathcal{Z}}(u, v) := 0$ for $u \ne v$, and $d_{\mathcal{Z}}(v, v) = 0$. Then $(\mathcal{Z}, d_{\mathcal{Z}})$ is a compact metric space, and in the setting specified in the statement of theorem 2 where **A3** is strengthened to independence, $s = p$ and $\mathbf{S}(v) = \sigma \mathbf{I}_p$ for all $v \in \mathcal{Z}$, the variables $\mathbf{Y}_1 \dots, \mathbf{Y}_n$; $\{\mathbf{X}(v), v \in \mathcal{Z}\}$; $\mathbf{E}_1, \dots, \mathbf{E}_n$ exactly follow the Latent Metric Model of Whiteley et al. [49].

Moreover, according to the description in section 2.1, the variables $Z_1, \dots, Z_n$ are i.i.d. according to a probability distribution supported on $\mathcal{Z}$. As in [49], by Mercer's theorem there exists a feature map $\phi : \mathcal{Z} \to \mathbb{R}^r$ associated with this probability distribution, such that $\langle \phi(u), \phi(v) \rangle = \alpha(u, v)$, for $u, v \in \mathcal{Z}$. Here $r$, as in **A6**, is the rank of the matrix with elements $\alpha(u, v)$, which is at most $\mathcal{Z}$.

Theorem 1 of [49] in this context implies there exists a random orthogonal matrix $\mathbf{Q} \in \mathbb{R}^{r \times r}$ such that

$$\max_{i \in [n]} \left\|p^{-1/2} \mathbf{Q} \zeta_i - \phi(Z_i)\right\| \in O_{\mathbb{P}}\left(\sqrt{\frac{nr}{p}} + \sqrt{\frac{r}{n}}\right). \quad (19)$$

Consider the bound:

$$\begin{aligned}
|\hat{\alpha}_{\text{pca}}(i, j) - \alpha(Z_i, Z_j)| &= \left|\frac{1}{p} \langle \zeta_i, \zeta_j \rangle - \langle \phi(Z_i), \phi(Z_j) \rangle\right| \\
&\le \left|\left\langle p^{-1/2} \mathbf{Q} \zeta_i - \phi(Z_i), p^{-1/2} \mathbf{Q} \zeta_j \right\rangle\right| \\
&\quad + \left|\left\langle \phi(Z_i), p^{-1/2} \mathbf{Q} \zeta_j - \phi(Z_j) \right\rangle\right| \\
&\le \left\|p^{-1/2} \mathbf{Q} \zeta_i - \phi(Z_i)\right\| \left(\left\|p^{-1/2} \mathbf{Q} \zeta_j - \phi(Z_j)\right\| + \|\phi(Z_j)\|\right) \\
&\quad + \|\phi(Z_i)\| \left\|p^{-1/2} \mathbf{Q} \zeta_i - \phi(Z_i)\right\|,
\end{aligned}$$

where orthogonality of $\mathbf{Q}$ has been used, and the final inequality uses Cauchy-Schwarz and the triangle inequality for the $\|\cdot\|$ norm. Combining the above estimate with (19), the bound:

$$\max_{i \in [n]} \|\phi(Z_i)\|^2 \le \max_{v \in \mathcal{Z}} \|\phi(v)\|^2 = \max_{v \in \mathcal{Z}} \alpha(v, v)$$

$$\le \sup_{j \ge 1} \max_{v \in \mathcal{Z}} \mathbb{E}[|X_j(v)|^2] \le \sup_{j \ge 1} \max_{v \in \mathcal{Z}} \mathbb{E}[|X_j(v)|^{2q}]^{1/q} \quad (20)$$

and **A4** completes the proof of the second claim of the theorem.

$\square$

**Proposition 2.** *Assume the model in section 2.1 satisfies assumptions A3-A5, and let $\varphi$ and $q$ be as in A3 and A4. Then there exists a constant $C(q, \varphi)$ depending only on $q$ and $\varphi$ such that for any $\delta > 0$,*

$$\mathbb{P}\left(\max_{1 \le i < j \le n} \left|p^{-1} \langle \mathbf{Y}_i, \mathbf{Y}_j \rangle - \alpha(Z_i, Z_j)\right| \ge \delta \,\middle|\, Z_1, \dots, Z_n\right)$$

$$\le \frac{1}{\delta^q} \frac{1}{p^{q/2}} \frac{n(n-1)}{2} C(q, \varphi) M(q, \mathbf{X}, \mathbf{E}, \mathbf{S}) \quad (21)$$

*where*

$$M(q, \mathbf{X}, \mathbf{E}, \mathbf{S}) := \sup_{k \geq 1} \max_{v \in \mathcal{Z}} \mathbb{E}\left[|X_k(v)|^{2q}\right]$$

$$+ \mathbb{E}\left[|\mathbf{E}_{11}|^q\right] \left(\sup_{p \geq 1} \max_{v \in \mathcal{Z}} \|\mathbf{S}(v)\|_{\mathrm{op}}^q\right) \sup_{k \geq 1} \max_{v \in \mathcal{Z}} \mathbb{E}\left[|X_k(v)|^q\right]$$

$$+ \mathbb{E}\left[|\mathbf{E}_{11}|^{2q}\right] \sup_{p \geq 1} \max_{v \in \mathcal{Z}} \|\mathbf{S}(v)\|_{\mathrm{op}}^{2q}.$$

*Proof.* Fix any $i, j$ such that $1 \leq i < j \leq n$. Consider the decomposition:

$$p^{-1} \langle \mathbf{Y}_i, \mathbf{Y}_j \rangle - \alpha(Z_i, Z_j) = \sum_{k=1}^{4} \Delta_k$$

where

$$\Delta_1 := p^{-1} \langle \mathbf{X}(Z_i), \mathbf{X}(Z_j) \rangle - \alpha(Z_i, Z_j)$$
$$\Delta_2 := p^{-1} \langle \mathbf{X}(Z_i), \mathbf{S}(Z_j)\mathbf{E}_j \rangle$$
$$\Delta_3 := p^{-1} \langle \mathbf{X}(Z_j), \mathbf{S}(Z_i)\mathbf{E}_i \rangle$$
$$\Delta_4 := p^{-1} \langle \mathbf{S}(Z_i)\mathbf{E}_i, \mathbf{S}(Z_j)\mathbf{E}_j \rangle$$

The proof proceeds by bounding $\mathbb{E}[|\Delta_k|^q | Z_1, \ldots, Z_n]$ for $k = 1, \ldots, 4$. Writing $\Delta_1$ as

$$\Delta_1 = \frac{1}{p} \sum_{k=1}^{p} \Delta_{1,k}, \qquad \Delta_{1,k} := X_k(Z_i)X_k(Z_j) - \mathbb{E}\left[X_k(Z_i)X_k(Z_j)| Z_1, \ldots, Z_n\right].$$

we see that $\Delta_1$ is a sum $p$ random variables each of which is conditionally mean zero given $Z_1, \ldots, Z_n$. Noting that the two collections of random variables $\{Z_1, \ldots, Z_n\}$ and $\{\mathbf{X}(v); v \in \mathcal{V}\}$ are independent (as per the description of the model in section 2.1), under assumption **A3** we may apply a moment inequality for $\varphi$-mixing random variables [51][Lemma 1.7] to show that there exists a constant $C_1(q, \varphi)$ depending only on $q, \varphi$ such that

$$\mathbb{E}\left[|\Delta_1|^q | Z_1, \ldots, Z_n\right]$$

$$\leq C_1(q, \varphi) \left\{ \frac{1}{p^q} \sum_{k=1}^{p} \mathbb{E}\left[|\Delta_{1,k}|^q | Z_1, \ldots, Z_n\right] + \left(\frac{1}{p^2} \sum_{k=1}^{p} \mathbb{E}\left[|\Delta_{1,k}|^2 | Z_1, \ldots, Z_n\right]\right)^{q/2} \right\}$$

$$\leq C_1(q, \varphi) \left\{ \frac{1}{p^q} \sum_{k=1}^{p} \mathbb{E}\left[|\Delta_{1,k}|^q | Z_1, \ldots, Z_n\right] + \frac{1}{p^{q/2}} \frac{1}{p} \sum_{k=1}^{p} \mathbb{E}\left[|\Delta_{1,k}|^q | Z_1, \ldots, Z_n\right] \right\}$$

$$\leq 2C_1(q, \varphi) \frac{1}{p^{q/2}} \sup_{k \geq 1} \mathbb{E}\left[|\Delta_{1,k}|^q | Z_1, \ldots, Z_n\right]$$

$$\leq 2^{q+1} C_1(q, \varphi) \frac{1}{p^{q/2}} \sup_{k \geq 1} \max_{v \in \mathcal{Z}} \mathbb{E}\left[|X_k(v)|^{2q}\right], \tag{22}$$

where second inequality holds by two applications of Jensen's inequality and $q \geq 2$, and the final inequality uses the fact that for $a, b \geq 0$, $(a + b)^q \leq 2^{q-1}(a^q + b^q)$, the Cauchy-Schwartz inequality, and the independence of $\{Z_1, \ldots, Z_n\}$ and $\{\mathbf{X}(v); v \in \mathcal{V}\}$.

For $\Delta_2$, we have

$$\Delta_2 := \frac{1}{p} \sum_{k=1}^{p} \Delta_{2,k}, \qquad \Delta_{2,k} := [\mathbf{S}(Z_j)^\top \mathbf{X}(Z_i)]_k \mathbf{E}_{jk},$$

where $[\cdot]_k$ denotes the $k$th element of a vector. Since the three collections of random variables, $\{Z_1, \ldots, Z_n\}$, $\{\mathbf{X}(v); v \in \mathcal{Z}\}$ and $\{\mathbf{E}_1, \ldots, \mathbf{E}_n\}$ are mutually independent, and the elements of each vector $\mathbf{E}_j \in \mathbb{R}^p$ are mean zero and independent, we see that given $\{Z_1, \ldots, Z_n\}$ and $\{\mathbf{X}(v); v \in \mathcal{V}\}$, $\Delta_2$ is a simple average of conditionally independent and conditionally mean-zero

random variables. Applying the Marcinkiewicz–Zygmund inequality we find there exists a constant $C_2(q)$ depending only on $q$ such that

$$\mathbb{E}\left[|\Delta_2|^q| \, Z_1,\ldots,Z_n,\mathbf{X}(v); v \in \mathcal{Z}\right]$$

$$\leq C_2(q)\mathbb{E}\left[\left|\frac{1}{p^2}\sum_{k=1}^{p}|\Delta_{2,k}|^2\right|^{q/2}\middle| \, Z_1,\ldots,Z_n,\mathbf{X}(v); v \in \mathcal{Z}\right]. \quad (23)$$

Noting that $q \geq 2$ and applying Minkowski's inequality to the r.h.s. of (23), then using the independence of $\{Z_1,\ldots,Z_n\}$, $\{\mathbf{X}(v); v \in \mathcal{Z}\}$ and $\{\mathbf{E}_1,\ldots,\mathbf{E}_n\}$ and the i.i.d. nature of the elements of the vector $\mathbf{E}_j$,

$$\mathbb{E}\left[\left|\frac{1}{p^2}\sum_{k=1}^{p}|\Delta_{2,k}|^2\right|^{q/2}\middle| \, Z_1,\ldots,Z_n,\mathbf{X}(v); v \in \mathcal{Z}\right]^{2/q}$$

$$\leq \frac{1}{p^2}\sum_{k=1}^{p}\mathbb{E}\left[|\Delta_{2,k}|^q| \, Z_1,\ldots,Z_n,\mathbf{X}(v); v \in \mathcal{Z}\right]^{2/q}$$

$$= \frac{1}{p^2}\sum_{k=1}^{p}\mathbb{E}\left[\left|[\mathbf{S}(Z_j)^\top \mathbf{X}(Z_i)]_k\right|^q |\mathbf{E}_{jk}|^q\middle| \, Z_1,\ldots,Z_n,\mathbf{X}(v); v \in \mathcal{Z}\right]^{2/q}$$

$$= \frac{1}{p^2}\mathbb{E}\left[|\mathbf{E}_{11}|^q\right]^{2/q}\sum_{k=1}^{p}\left|[\mathbf{S}(Z_j)^\top \mathbf{X}(Z_i)]_k\right|^2$$

$$= \frac{1}{p^2}\mathbb{E}\left[|\mathbf{E}_{11}|^q\right]^{2/q}\left\|\mathbf{S}(Z_j)^\top \mathbf{X}(Z_i)\right\|^2$$

$$\leq \frac{1}{p^2}\mathbb{E}\left[|\mathbf{E}_{11}|^q\right]^{2/q}\max_{v \in \mathcal{Z}}\|\mathbf{S}(v)\|_{\text{op}}^2\|\mathbf{X}(Z_i)\|^2.$$

Substituting into (23) and using the tower property of conditional expectation we obtain:

$$\mathbb{E}\left[|\Delta_2|^q| \, Z_1,\ldots,Z_n\right]$$

$$\leq \frac{1}{p^{q/2}}\mathbb{E}\left[|\mathbf{E}_{11}|^q\right]\max_{v \in \mathcal{Z}}\|\mathbf{S}(v)\|_{\text{op}}^q\frac{1}{p^{q/2}}\mathbb{E}\left[\|\mathbf{X}(Z_i)\|^q| \, Z_1,\ldots,Z_n\right]$$

$$= \frac{1}{p^{q/2}}\mathbb{E}\left[|\mathbf{E}_{11}|^q\right]\max_{v \in \mathcal{Z}}\|\mathbf{S}(v)\|_{\text{op}}^q\mathbb{E}\left[\left(\frac{1}{p}\sum_{k=1}^{p}|X_k(Z_j)|^2\right)^{q/2}\middle| \, Z_1,\ldots,Z_n\right]$$

$$\leq \frac{1}{p^{q/2}}\mathbb{E}\left[|\mathbf{E}_{11}|^q\right]\max_{v \in \mathcal{Z}}\|\mathbf{S}(v)\|_{\text{op}}^q\mathbb{E}\left[\frac{1}{p}\sum_{k=1}^{p}|X_k(Z_j)|^q\middle| \, Z_1,\ldots,Z_n\right]$$

$$\leq \frac{1}{p^{q/2}}\mathbb{E}\left[|\mathbf{E}_{11}|^q\right]\max_{v \in \mathcal{Z}}\|\mathbf{S}(v)\|_{\text{op}}^q\sup_{k \geq 1}\max_{v \in \mathcal{Z}}\mathbb{E}\left[|X_k(v)|^q\right] \quad (24)$$

where the second inequality holds by Jensen's inequality (recall $q \geq 2$). Since the r.h.s. of (24) does not depend on $i$ or $j$, the same bound holds with $\Delta_2$ on the l.h.s. replaced by $\Delta_3$.

Turning to $\Delta_4$, we have

$$\Delta_4 := \frac{1}{p}\langle\mathbf{S}(Z_i)\mathbf{E}_i, \mathbf{S}(Z_j)\mathbf{E}_j\rangle = \frac{1}{p}\sum_{1 \leq k,\ell \leq p}\Delta_{4,k,\ell}, \qquad \Delta_{4,k,\ell} := \mathbf{E}_{ik}\mathbf{E}_{j\ell}[\mathbf{S}(Z_i)^\top\mathbf{S}(Z_j)]_{k\ell}.$$

Noting that $i \neq j$, and that the elements of $\mathbf{E}_i$ and $\mathbf{E}_j$ are independent, identically distributed, and mean zero, we see that $\Delta_4$ is a sum of $p^2$ random variables which are all conditionally mean zero and conditionally independent given $Z_1,\ldots,Z_n$. The Marcinkiewicz–Zygmund inequality gives:

$$\mathbb{E}\left[|\Delta_4|^q| \, Z_1,\ldots,Z_n\right] \leq C_2(q)\mathbb{E}\left[\left|\frac{1}{p^2}\sum_{1 \leq k,\ell \leq s}|\Delta_{4,k,\ell}|^2\right|^{q/2}\middle| \, Z_1,\ldots,Z_n\right]. \quad (25)$$

Applying Minkowski's inequality to the r.h.s. of (25),

$$\mathbb{E}\left[\left|\left|\frac{1}{p^2}\sum_{1\le k,\ell\le p}|\Delta_{4,k,\ell}|^2\right|^{q/2}\right|\,Z_1,\ldots,Z_n\right]^{2/q}$$

$$\le\frac{1}{p^2}\sum_{1\le k,\ell\le p}\mathbb{E}\left[|\Delta_{4,k,\ell}|^q|\,Z_1,\ldots,Z_n\right]^{2/q}$$

$$=\frac{1}{p^2}\sum_{1\le k,\ell\le p}\mathbb{E}\left[|\mathbf{E}_{ik}|^q\,|\mathbf{E}_{j\ell}|^q\,\left|[\mathbf{S}(Z_i)^\top\mathbf{S}(Z_j)]_{k\ell}\right|^q\,\Big|\,Z_1,\ldots,Z_n\right]^{2/q}$$

$$=\frac{1}{p^2}\mathbb{E}\left[|\mathbf{E}_{11}|^q\right]^{4/q}\sum_{1\le k,\ell\le p}\left|[\mathbf{S}(Z_i)^\top\mathbf{S}(Z_j)]_{k\ell}\right|^2$$

$$\le\frac{1}{p^2}\mathbb{E}\left[|\mathbf{E}_{11}|^{2q}\right]^{2/q}\max_{u,v\in\mathcal{Z}}\|\mathbf{S}(u)^\top\mathbf{S}(v)\|_{\mathrm{F}}^2,$$

where the final inequality holds by Jensen's inequality. Substituting back into (25) and using $\|\mathbf{S}(u)^\top\mathbf{S}(v)\|_{\mathrm{F}}\le p^{1/2}\|\mathbf{S}(u)^\top\mathbf{S}(v)\|_{\mathrm{op}}\le p^{1/2}\|\mathbf{S}(u)\|_{\mathrm{op}}\|\mathbf{S}(v)\|_{\mathrm{op}}$, we obtain:

$$\mathbb{E}\left[|\Delta_4|^q|\,Z_1,\ldots,Z_n\right]\le C_2(q)\frac{1}{p^{q/2}}\mathbb{E}\left[|\mathbf{E}_{11}|^{2q}\right]\max_{u\in\mathcal{Z}}\|\mathbf{S}(u)\|_{\mathrm{op}}^{2q}.\tag{26}$$

Combining (22), (24) and (26) using the fact that for $a,b\ge 0$, $(a+b)^q\le 2^{q-1}(a^q+b^q)$, we find that there exists a constant $C(q,\varphi)$ depending only on $q$ and $\varphi$ such that

$$\mathbb{E}\left[\left|p^{-1}\langle\mathbf{Y}_i,\mathbf{Y}_j\rangle-\alpha(Z_i,Z_j)\right|^q\,\Big|\,Z_1,\ldots,Z_n\right]$$

$$\le C(q,\varphi)\frac{1}{p^{q/2}}M(q,\mathbf{X},\mathbf{E},\mathbf{S}),$$

where $M(q,\mathbf{X},\mathbf{E},\mathbf{S})$ is defined in the statement of the proposition and is finite by assumptions **A4** and **A5**. By Markov's inequality, for any $\delta\ge 0$,

$$\mathbb{P}\left(\left|p^{-1}\langle\mathbf{Y}_i,\mathbf{Y}_j\rangle-\alpha(Z_i,Z_j)\right|\ge\delta\big|\,Z_1,\ldots,Z_n\right)\le\frac{1}{\delta^q}C(q,\varphi)\frac{1}{p^{q/2}}M(q,\mathbf{X},\mathbf{E},\mathbf{S})\tag{27}$$

and the proof is completed by a union bound:

$$\mathbb{P}\left(\max_{1\le i<j\le n}\left|p^{-1}\langle\mathbf{Y}_i,\mathbf{Y}_j\rangle-\alpha(Z_i,Z_j)\right|<\delta\Big|\,Z_1,\ldots,Z_n\right)$$

$$=\mathbb{P}\left(\bigcap_{1\le i<j\le n}\left|p^{-1}\langle\mathbf{Y}_i,\mathbf{Y}_j\rangle-\alpha(Z_i,Z_j)\right|<\delta\Big|\,Z_1,\ldots,Z_n\right)$$

$$=1-\mathbb{P}\left(\bigcup_{1\le i<j\le n}\left|p^{-1}\langle\mathbf{Y}_i,\mathbf{Y}_j\rangle-\alpha(Z_i,Z_j)\right|\ge\delta\Big|\,Z_1,\ldots,Z_n\right)$$

$$\ge 1-\frac{n(n-1)}{2}\frac{1}{\delta^q}C(q,\varphi)\frac{1}{p^{q/2}}M(q,\mathbf{X},\mathbf{E},\mathbf{S}).$$

$\square$

### C.3 Interpretation of merge heights and exact tree recovery

Here we expand on the discussion in section 3.3 and provide further interpretation of merge heights and algorithm 1. In particular our aim is to clarify in what circumstances algorithm 1 will asymptotically correctly recover underlying tree structure. For ease of exposition throughout section C.3 we assume that $\tilde{\mathcal{Z}}$ are the leaf vertices of $\mathcal{T}$.

As a preliminary we note the following corollary to theorem 1: assuming $b>0$, if one takes as input to algorithm 1 the true merge heights, i.e. (up to bijective relabelling of leaf vertices)

$\hat{\alpha}(\cdot, \cdot) := m(\cdot, \cdot) = \alpha(\cdot, \cdot)$, where $n = |\mathcal{Z}|$, then theorem 1 implies that algorithm 1 outputs a dendrogram $\mathcal{D}$ whose merge heights $\hat{m}(\cdot, \cdot)$ are equal to $m(\cdot, \cdot)$ (up to bijective relabeling over vertices). This clarifies that with knowledge of $m(\cdot, \cdot)$, algorithm 1) constructs a dendrogram which has $m(\cdot, \cdot)$ as its merge heights.

We now ask for more: if once again $m(\cdot, \cdot)$ is taken as input to algorithm 1, under what conditions is the output tree $\hat{\mathcal{T}}$ equal to $\mathcal{T}$ (upto bijective relabelling of vertices)? We claim this holds when $\mathcal{T}$ is a binary tree and that all its non-leaf nodes have different heights. We provide a sketch proof of this claim, since a complete proof involves many tedious and notationally cumbersome details.

To remove the need for repeated considerations of relabelling, suppose $\mathcal{T} = (\mathcal{V}, \mathcal{E})$ is given, then w.l.o.g. relabel the leaf vertices of $\mathcal{T}$ as $\{1\}, \ldots, \{|\mathcal{Z}|\}$ and relabel each non-leaf vertex to be the union of its children. Thus each vertex is some subset of $[|\mathcal{Z}|]$.

Now assume that $\mathcal{T}$ is a binary tree and that all its non-leaf nodes have different heights. Note that $|\mathcal{V}| = 2|\mathcal{Z}| - 1$, i.e., there are $|\mathcal{Z}| - 1$ non-leaf vertices. The tree $\mathcal{T}$ is uniquely characterized by a sequence of partitions $\tilde{P}_0, \ldots, \tilde{P}_{|\mathcal{Z}|-1}$ where $\tilde{P}_0 := \{\{1\}, \ldots, \{|\mathcal{Z}|\}\}$, and for $m = 1, \ldots, |\mathcal{Z}| - 1$, $\tilde{P}_m$ is constructed from $\tilde{P}_{m-1}$ by merging the two elements of $\tilde{P}_{m-1}$ whose most recent common ancestor is the $m$th highest non-leaf vertex (which is uniquely defined since we are assuming no two non-leaf vertices have equal heights).

To see that in this situation algorithm 1, with $\hat{\alpha}(\cdot, \cdot) := m(\cdot, \cdot)$ and $n = |\mathcal{Z}|$ as input, performs exact recovery of the tree, i.e., $\hat{\mathcal{T}} = \mathcal{T}$, it suffices to notice that the sequence of partitions $P_0, \ldots, P_{|\mathcal{Z}|-1}$ constructed by algorithm 1 uniquely characterizes $\hat{\mathcal{T}}$, and moreover $(\tilde{P}_0, \ldots, \tilde{P}_{|\mathcal{Z}|-1}) = (P_0, \ldots, P_{|\mathcal{Z}|-1})$. The details of this last equality involve simple but tedious substitutions of $m(\cdot, \cdot)$ in place of $\hat{\alpha}(\cdot, \cdot)$ in algorithm 1, so are omitted.

# D   Further details of numerical experiments and data preparation

All real datasets used are publicly available under the CC0: Public domain license. Further, all experiments were run locally on a laptop with an integrated GPU (Intel UHD Graphics 620).

## D.1   Simulated data

For each $v \in \mathcal{V}$, $X_1(v), \ldots, X_p(v)$ are independent and identically distributed Gaussian random variables with:

$$
\begin{aligned}
X_j(1) &\sim N(X_j(6), 5), \\
X_j(2) &\sim N(X_j(6), 2), \\
X_j(3) &\sim N(X_j(6), 2), \\
X_j(4) &\sim N(X_j(7), 0.5), \\
X_j(5) &\sim N(X_j(7), 7), \\
X_j(6) &\sim N(X_j(8), 2), \\
X_j(7) &\sim N(X_j(8), 1), \\
X_j(8) &\sim N(0, 1),
\end{aligned}
$$

for $j = 1, \ldots, p$.

## D.2   20 Newsgroups

The dataset originates from [29], however, the version used is the one available in the Python package 'scikit-learn' [38]. Each document is pre-processed in the following way: generic stopwords and e-mail addresses are removed, and words are lemmatised. The processed documents are then converted into a matrix of TF-IDF features. Labels can be found on the 20 Newsgroups website `http://qwone.com/~jason/20Newsgroups/`, but are mainly intuitive from the title of labels, with full stops separating levels of hierarchy. When using PCA a dimension of $r = 34$ was selected by the method described in appendix A.

The following numerical results complement those in the main part of the paper.

Table 2: Kendall $\tau_b$ ranking performance measure, for Algorithm 1 and the 20 Newsgroups data set. The mean Kendall $\tau_b$ correlation coefficient is reported alongside the standard error (numerical value shown is the standard error$\times 10^3$). This numerical results are plotted in figure 4 below.

| Data | Input | $p = 500$ | $p = 1000$ | $p = 5000$ | $p = 7500$ | $p = 12818$ |
|---|---|---|---|---|---|---|
| Newsgroups | $\mathbf{Y}_{1:n}$ | 0.026 (0.55) | 0.016 (1.0) | 0.13 (2.2) | 0.17 (2.5) | 0.26 (2.9) |
| | $\zeta_{1:n}$ | 0.017 (0.72) | 0.047 (1.2) | 0.12 (1.9) | 0.15 (2.5) | 0.24 (2.6) |

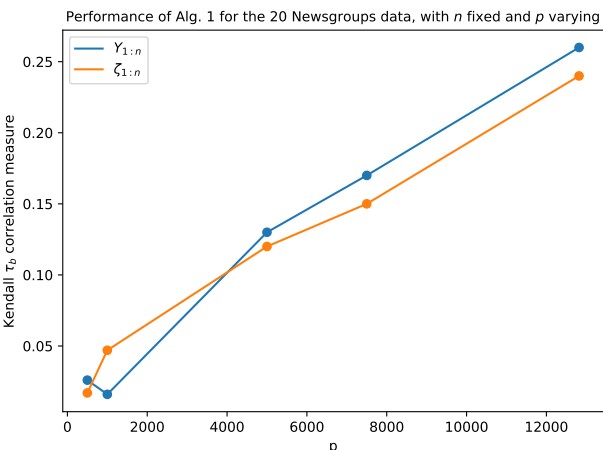

Figure 4: Performance of Algorithm 1 for the 20 Newsgroups data set as a function of number of TF-IDF features, $p$, with $n$ fixed. See table 1 for numerical values and standard errors.

### D.3 Zebrafish gene counts

These data were collected over a 24-hour period (timestamps available in the data), however, the temporal aspect of the data was ignored when selecting a sub-sample. To process the data, we followed the steps in [47] which are the standard steps in the popular SCANPY [50] – a Python package for analysing single-cell gene-expression data to process the data. This involves filtering out genes that are present in less than 3 cells or are highly variable, taking the logarithm, scaling and regressing out the effects of the total counts per cell. When using PCA a dimension of $r = 29$ was selected by the method described in appendix A.

### D.4 Amazon reviews

The pre-processing of this dataset is similar to that of the Newsgroup data except e-mail addresses are no longer removed. Some data points had a label of 'unknown' in the third level of the hierarchy, these were removed from the dataset. In addition, reviews that are two words or less are not included. When using PCA a dimension of $r = 22$ was selected by the method described in appendix A.

### D.5 S&P 500 stock data

This dataset was found through the paper [8] with the authors code used to process the data. The labels follow the Global Industry Classification Standard and can be found here: [3]. The return on day $i$ of each stock is calculated by

$$\text{return}_i = \frac{p_i^{cl} - p_i^{op}}{p_i^{op}},$$

where $p_i^{op}$ and $p_i^{cl}$ is the respective opening and closing price on day $i$. When using PCA a dimension of $r = 10$ was used as selected by the method described in appendix A.

## D.6 Additional method comparison

Table 3 reports additional method comparison results, complementing those in table 1 which concerned only the average linkage function (noting UPGMA is equivalent to using the average linkage function with Euclidean distances). In table 3 we also compare to Euclidean and cosine distances paired with complete and single linkage functions. To aid comparison, the first column (average linkage with the dot product) is the same as in table 1. In general, using complete or single linkage performs worse for both Euclidean and cosine distances. The only notable exception being a slight improvement on the simulated dataset.

Table 3: Kendall $\tau_b$ ranking performance measure. For the dot product method, i.e., algorithm 1, $\mathbf{Y}_{1:n}$ as input corresponds to using $\hat{\alpha}_{\text{data}}$, and $\zeta_{1:n}$ corresponds to $\hat{\alpha}_{\text{pca}}$. The mean Kendall $\tau_b$ correlation coefficient is reported alongside the standard error (numerical value shown is the standard error $\times 10^3$).

| Data | Input | Average linkage
Dot product | Complete linkage
Euclidean | Complete linkage
Cosine | Single linkage
Euclidean | Single linkage
Cosine |
|------|-------|------|------|------|------|------|
| Newsgroups | $\mathbf{Y}_{1:n}$ | 0.26 (2.9) | 0.022 (0.87) | -0.010 (0.88) | -0.0025 (0.62) | -0.0025 (0.62) |
| | $\zeta_{1:n}$ | 0.24 (2.6) | 0.0041 (1.2) | 0.036 (1.2) | -0.016 (2.0) | 0.067 (1.5) |
| Zebrafish | $\mathbf{Y}_{1:n}$ | 0.34 (3.4) | 0.15 (2.2) | 0.24 (3.2) | 0.023 (3.0) | 0.032 (2.9) |
| | $\zeta_{1:n}$ | 0.34 (3.4) | 0.17 (2.0) | 0.30 (3.4) | 0.12 (2.8) | 0.15 (3.2) |
| Reviews | $\mathbf{Y}_{1:n}$ | 0.15 (2.5) | 0.019 (0.90) | 0.023 (1.0) | 0.0013 (0.81) | 0.0013 (0.81) |
| | $\zeta_{1:n}$ | 0.14 (2.4) | 0.058 (1.5) | 0.063 (1.8) | 0.015 (1.2) | 0.038 (1.0) |
| S&P 500 | $\mathbf{Y}_{1:n}$ | 0.34 (10) | 0.33 (10) | 0.33 (10) | 0.17 (10) | 0.17 (10) |
| | $\zeta_{1:n}$ | 0.36 (9.4) | 0.32 (10) | 0.31 (10) | 0.36 (13) | 0.39 (12) |
| Simulated | $\mathbf{Y}_{1:n}$ | 0.86 (1) | 0.55 (8.7) | 0.84 (2.0) | 0.55 (8.7) | 0.84 (2.0) |
| | $\zeta_{1:n}$ | 0.86 (1) | 0.55 (8.7) | 0.84 (2.0) | 0.55 (8.7) | 0.84 (2.0) |

Table 4: Kendall $\tau_b$ ranking performance measure. The mean Kendall $\tau_b$ correlation coefficient is reported alongside the standard error (numerical value shown is the standard error $\times 10^3$).

| Data | Input | UPGMA with dot product *dissimilarity* | UPGMA with Manhattan distance |
|------|-------|------|------|
| Newsgroups | $\mathbf{Y}_{1:n}$ | -0.0053 (0.24) | -0.0099 (1.3) |
| | $\zeta_{1:n}$ | 0.0029 (0.33) | 0.052 (1.6) |
| Zebrafish | $\mathbf{Y}_{1:n}$ | 0.0012 (0.13) | 0.16 (2.4) |
| | $\zeta_{1:n}$ | 0.00046 (0.12) | 0.050 (2.8) |
| Reviews | $\mathbf{Y}_{1:n}$ | -0.0005 (0.29) | 0.0018 (0.44) |
| | $\zeta_{1:n}$ | -0.0015 (0.41) | 0.061 (1.3) |
| S&P 500 | $\mathbf{Y}_{1:n}$ | 0.0026 (7.7) | 0.37 (9.4) |
| | $\zeta_{1:n}$ | 0.0028 (7.5) | 0.39 (11) |
| Simulated | $\mathbf{Y}_{1:n}$ | -0.0026 (1.6) | 0.55 (8.7) |
| | $\zeta_{1:n}$ | -0.0023 (1.8) | 0.84 (2) |

# E Understanding agglomerative clustering with Euclidean or cosine distances in our framework

## E.1 Quantifying dissimilarity using Euclidean distance

The first step of many standard variants of agglomerative clustering such as UPGMA and Ward's method is to find and merge the pair of data vectors which are closest to each other in Euclidean distance. From the elementary identity:

$$\|\mathbf{Y}_i - \mathbf{Y}_j\|^2 = \|\mathbf{Y}_i\|^2 + \|\mathbf{Y}_j\|^2 - 2\langle \mathbf{Y}_i, \mathbf{Y}_j \rangle,$$

we see that, in general, this is not equivalent to finding the pair with largest dot product, because of the presence of the terms $\|\mathbf{Y}_i\|^2$ and $\|\mathbf{Y}_j\|^2$. For some further insight in to how this relates to our

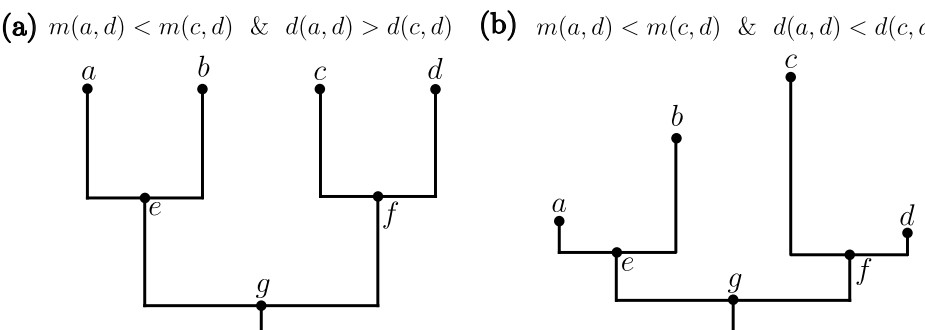

**(a)** $m(a,d) < m(c,d)$ & $d(a,d) > d(c,d)$    **(b)** $m(a,d) < m(c,d)$ & $d(a,d) < d(c,d)$

Figure 5: Illustration of how maximising merge height $m(\cdot,\cdot)$ may or may not be equivalent to minimising distance $d(\cdot,\cdot)$, depending on the geometry of the dendrogram. (a) Equivalence holds (b) Equivalence does not hold.

modelling and theoretical analysis framework, it is revealing to the consider the idealised case of choosing to merge by maximising merge height $m(\cdot,\cdot)$ versus minimising $d(\cdot,\cdot)$ (recall the identities for $m$ and $d$ established in lemma 1). Figure 5 shows two simple scenarios in which geometry of the dendrogram has an impact on whether or not maximising merge height $m(\cdot,\cdot)$ is equivalent to minimising $d(\cdot,\cdot)$. From this example we see that, in situations where some branch lengths are disproportionately large, minimising $d(\cdot,\cdot)$ will have different results to maximising $m(\cdot,\cdot)$.

UPGMA and Ward's method differ in their linkage functions, and so differ in the clusters they create in practice after their respective first algorithmic steps. UPGMA uses average linkage to combine Euclidean distances, and there does not seem to be a mathematically simple connection between this and algorithm 1, except to say that in general it will return different results. Ward's method merges the pair of clusters which results in the minimum increase in within-cluster variance. When clusters contain equal numbers of samples, this increase is equal to the squared Euclidean distance between the clusters' respective centroids.

### E.2 Agglomerative clustering with cosine distance is equivalent to an instance of algorithm 1

The cosine 'distance' between $\mathbf{Y}_i$ and $\mathbf{Y}_j$ is:

$$d_{\cos}(i,j) := 1 - \frac{\langle \mathbf{Y}_i, \mathbf{Y}_j \rangle}{\|\mathbf{Y}_i\|\|\mathbf{Y}_j\|}.$$

In table 1 we show results for standard agglomerative clustering using $d_{\cos}$ as a measure of dissimilarity, combined with average linkage. At each iteration, this algorithm works by merging the pair of data-vectors/clusters which are closest with respect to $d_{\cos}(\cdot,\cdot)$, say $u$ and $v$ merged to form $w$, with dissimilarities between $w$ and the existing data-vectors/clusters computed according to the average linkage function:

$$d_{\cos}(w,\cdot) := \frac{|u|}{|w|}d_{\cos}(u,\cdot) + \frac{|v|}{|w|}d_{\cos}(v,\cdot). \tag{28}$$

This procedure can be seen to be equivalent to algorithm 1 with input $\hat{\alpha}(\cdot,\cdot) = 1 - d_{\cos}(\cdot,\cdot)$. Indeed maximising $1 - d_{\cos}(\cdot,\cdot)$ is clearly equivalent to minimizing $d_{\cos}(\cdot,\cdot)$, and with $\hat{\alpha}(\cdot,\cdot) = 1 - d_{\cos}(\cdot,\cdot)$ the affinity computation at line 6 of algorithm 1 is:

$$\hat{\alpha}(w,\cdot) := \frac{|u|}{|w|}\hat{\alpha}(u,\cdot) + \frac{|v|}{|w|}\hat{\alpha}(v,\cdot)$$

$$= \frac{|u|}{|w|}\left[1 - d_{\cos}(u,\cdot)\right] + \frac{|v|}{|w|}\left[1 - d_{\cos}(v,\cdot)\right]$$

$$= \frac{|u| + |v|}{|w|} - \frac{|u|}{|w|}d_{\cos}(u,\cdot) - \frac{|u|}{|w|}d_{\cos}(v,\cdot)$$

$$= 1 - \left[\frac{|u|}{|w|}d_{\cos}(u,\cdot) + \frac{|u|}{|w|}d_{\cos}(v,\cdot)\right],$$

where on the r.h.s. of the final equality we recognise (28).

### E.3 Using cosine similarity as an affinity measure removes multiplicative noise

Building from the algorithmic equivalence identified in section E.2, we now address the theoretical performance of agglomerative clustering with cosine distance in our modelling framework.

Intuitively, cosine distance is used in situations where the magnitudes of data vectors are thought not to convey useful information about dissimilarity. To formalise this idea, we consider a variation of our statistical model from section 2.1 in which:

- $\mathcal{Z}$ are the leaf vertices of $\mathcal{T}$, $|\mathcal{Z}| = n$, and we take these vertices to be labelled $\mathcal{Z} = \{1, \ldots, n\}$
- $\mathbf{X}$ is as in section 2.1, properties **A1** and **A2** hold, and it is assumed that for all $v \in \mathcal{Z}$, $p^{-1}\mathbb{E}[\|\mathbf{X}(v)\|^2] = 1$.
- the additive model (1) is replaced by a multiplicative noise model:
$$\mathbf{Y}_i = \gamma_i \mathbf{X}(i), \tag{29}$$
where $\gamma_i > 0$ are all strictly positive random scalars, independent of other variables, but otherwise arbitrary.

The interpretation of this model is that the expected square magnitude of the data vector $\mathbf{Y}_i$ is entirely determined by $\gamma_i$, indeed we have
$$\frac{1}{p}\mathbb{E}[\|\mathbf{Y}_i\|^2|\gamma_i] = \gamma_i^2 \frac{1}{p}\mathbb{E}[\|\mathbf{X}(i)\|^2] = \gamma_i^2 h(i) = \gamma_i^2,$$
where $h$ is as in section 2.1. We note that in this multiplicative model, the random vectors $\mathbf{E}_i$ and matrices $\mathbf{S}(v)$ from section 2.1 play no role, and one can view the random variables $Z_1, \ldots, Z_n$ as being replaced by constants $Z_i = i$, rather than being i.i.d.

Now define:
$$\hat{\alpha}_{\cos}(i,j) := \frac{\langle \mathbf{Y}_i, \mathbf{Y}_j \rangle}{\|\mathbf{Y}_i\|\|\mathbf{Y}_j\|} = 1 - d_{\cos}(i,j).$$

**Theorem 3.** *Assume that the model specified in section E.3 satisfies A3 and for some $q \geq 2$, $\sup_{j \geq 1} \max_{v \in \mathcal{Z}} \mathbb{E}[|X_j(v)|^{2q}] < \infty$. Then*
$$\max_{i,j \in [n], i \neq j} |\alpha(i,j) - \hat{\alpha}_{\cos}(i,j)| \in O_{\mathbb{P}}\left(\frac{n^{2/q}}{\sqrt{p}}\right).$$

*Proof.* The main ideas of the proof are that the multiplicative factors $\gamma_i, \gamma_j$ in the numerator $\hat{\alpha}_{\cos}(i,j)$ cancel out with those in the denominator, and combined with the condition $p^{-1}\mathbb{E}[\|\mathbf{X}(v)\|^2] = 1$ we may then establish that $\alpha(i,j)$ and $\hat{\alpha}_{\cos}(i,j)$ are probabilistically close using similar arguments to those in the proof of proposition 2.

Consider the decomposition:
$$\alpha(i,j) - \hat{\alpha}_{\cos}(i,j) = \alpha(i,j) - \frac{p^{-1}\langle \mathbf{X}(i), \mathbf{X}(j) \rangle}{p^{-1/2}\|\mathbf{X}(i)\| p^{-1/2}\|\mathbf{X}(j)\|}$$
$$= \alpha(i,j) - \frac{1}{p}\langle \mathbf{X}(i), \mathbf{X}(j) \rangle \tag{30}$$
$$+ \frac{\frac{1}{p}\langle \mathbf{X}(i), \mathbf{X}(j) \rangle}{p^{-1/2}\|\mathbf{X}(i)\| p^{-1/2}\|\mathbf{X}(j)\|}\left[p^{-1/2}\|\mathbf{X}(i)\| p^{-1/2}\|\mathbf{X}(j)\| - 1\right]. \tag{31}$$

Applying the Cauchy-Schwartz inequality, and adding and subtracting $p^{-1/2}\|\mathbf{X}(j)\|$ and $1$ in the final term of this decomposition leads to:
$$\max_{i,j \in [n], i \neq j} |\alpha(i,j) - \hat{\alpha}_{\cos}(i,j)|$$
$$\leq \max_{i,j \in [n], i \neq j}\left|\alpha(i,j) - \frac{1}{p}\langle \mathbf{X}(i), \mathbf{X}(j) \rangle\right| \tag{32}$$
$$+ \max_{i \in [n]}\left|p^{-1/2}\|\mathbf{X}(i)\| - 1\right| \tag{33}$$
$$+ \left(1 + \max_{i \in [n]}\left|p^{-1/2}\|\mathbf{X}(i)\| - 1\right|\right)\max_{j \in [n]}\left|p^{-1/2}\|\mathbf{X}(j)\| - 1\right|. \tag{34}$$

The proof proceeds by arguing that the term (32) is in $O_{\mathbb{P}}\left(n^{2/q}/\sqrt{p}\right)$, and the terms (33) and (34) are in $O_{\mathbb{P}}\left(n^{1/q}/\sqrt{p}\right)$, where we note that this asymptotic concerns the limit as $n^{2/q}/\sqrt{p} \to 0$.

In order to analyse the term (32), let $\tilde{\mathbb{P}}$ denote the probability law of the additive model for $\mathbf{Y}_1, \ldots, \mathbf{Y}_n$ in equation (1) in section 2.1, in the case that $\mathbf{S}(v) = 0$ for all $v \in \mathcal{Z}$. Let $\tilde{\mathbb{E}}$ denote the associated expectation. Then for any $\delta > 0$ and $i, j \in [n]$, $i \neq j$,

$$
\begin{aligned}
&\mathbb{P}\left(\left|\alpha(i,j) - \frac{1}{p}\langle \mathbf{X}(i), \mathbf{X}(j)\rangle\right| > \delta\right) \\
&= \tilde{\mathbb{P}}\left(\left|\frac{1}{p}\langle \mathbf{Y}_i, \mathbf{Y}_j\rangle - \alpha(Z_i, Z_j)\right| > \delta \,\middle|\, Z_1 = 1, \ldots, Z_n = n\right) \\
&= \frac{\tilde{\mathbb{E}}\left[\mathbb{I}\{Z_1 = 1, \ldots, Z_n = n\}\tilde{\mathbb{P}}\left(\left|\frac{1}{p}\langle \mathbf{Y}_i, \mathbf{Y}_j\rangle - \alpha(Z_i, Z_j)\right| > \delta \,\middle|\, Z_1, \ldots, Z_n\right)\right]}{\tilde{\mathbb{P}}(\{Z_1 = 1, \ldots, Z_n = n\})}.
\end{aligned}
$$

The conditional probability on the r.h.s of the final equality can be upper bounded using the inequality (27) in the proof of proposition 2, and combined with the same union bound argument used immediately after (27), this establishes (32) is in $O_{\mathbb{P}}\left(n^{2/q}/\sqrt{p}\right)$ as required.

To show that (33) and (34) are in $O_{\mathbb{P}}\left(n^{1/q}/\sqrt{p}\right)$, it suffices to show that $\max_{i \in [n]}\left|p^{-1/2}\|\mathbf{X}(i)\| - 1\right|$ is in $O_{\mathbb{P}}\left(n^{1/q}/\sqrt{p}\right)$. By re-arranging the equality: $(|a| - 1)(|a| + 1) = |a|^2 - 1$, we have

$$
\left|p^{-1/2}\|\mathbf{X}(i)\| - 1\right| \leq \left|p^{-1}\|\mathbf{X}(i)\|^2 - 1\right| = \left|p^{-1}\sum_{j=1}^{p}\Delta_j(i)\right|, \tag{35}
$$

where $\Delta_j(i) := |X_j(i)|^2 - 1$. Thus under the model from section E.3, $p^{-1}\sum_{j=1}^{p}\Delta_j(i)$ is an average of $p$ mean-zero random variables, and by the same arguments as in the proof of proposition 2 under the mixing assumption **A3** and the assumption of the theorem that $\sup_{j \geq 1}\max_{v \in \mathcal{Z}}\mathbb{E}[|X_j(v)|^{2q}] < \infty$, combined with a union bound, we have

$$
\max_{i \in [n]}\left|p^{-1}\|\mathbf{X}(i)\|^2 - 1\right| \in O_{\mathbb{P}}(n^{1/q}/\sqrt{p}).
$$

Together with (35) this implies $\max_{i \in [n]}\left|p^{-1/2}\|\mathbf{X}(i)\| - 1\right|$ is in $O_{\mathbb{P}}\left(n^{1/q}/\sqrt{p}\right)$ as required, and that completes the proof.

$\square$

### E.4 Limitations of our modelling assumptions and failings of dot-product affinities

As noted in section 5, algorithm 1 is motivated and theoretically justified by our modelling assumptions, laid out in sections 2-3 and E.3. If these assumptions are not well matched to data in practice, then algorithm 1 may not perform well.

The proof of lemma 3 shows that, as a consequence of the conditional independence and martingale-like assumptions, **A1** and **A2**, $\alpha(u, v) \geq 0$ for all $u, v$. By theorems 2 and 3, $\hat{\alpha}_{\text{data}}$, $\hat{\alpha}_{\text{pca}}$ approximate $\alpha$ (at the vertices $\mathcal{Z}$) under the additive model from section 2.1, and $\hat{\alpha}_{\text{cos}}$ approximates $\alpha$ under the multiplicative model from section E.3. Therefore if in practice $\hat{\alpha}_{\text{data}}(i, j)$, $\hat{\alpha}_{\text{pca}}(i, j)$ or $\hat{\alpha}_{\text{cos}}(i, j)$ are found to take non-negligible negative values for some pairs $i, j$, that is an indication that our modelling assumptions may not be appropriate.

Even if the values taken by $\hat{\alpha}_{\text{data}}(i, j)$, $\hat{\alpha}_{\text{pca}}(i, j)$ or $\hat{\alpha}_{\text{cos}}(i, j)$ are all positive in practice, there could be other failings of our modelling assumptions. As an academic but revealing example, if instead of (1) or (29), the data were to actually follow a combined additive *and* multiplicative noise model, i.e.,

$$
\mathbf{Y}_i = \gamma_i \mathbf{X}(Z_i) + \mathbf{S}(Z_i)\mathbf{E}_i,
$$

then in general neither $\hat{\alpha}_{\text{data}}$, $\hat{\alpha}_{\text{pca}}$ nor $\hat{\alpha}_{\text{cos}}$ would approximate $\alpha$ as $p \to \infty$.

