# OpenReview forum: "Hierarchical clustering with dot products recovers hidden tree structure"
_NeurIPS.cc/2023/Conference — NeurIPS 2023 spotlight_

### Official Review · Reviewer_Emfa · 2023-07-03

**Soundness:** 4 excellent
**Presentation:** 4 excellent
**Contribution:** 3 good
**Rating:** 7
**Confidence:** 3

**Summary:**

For agglomerative clustering, this paper proposes a method where the affinity of clusters at each hierarchy is naturally visualized as their height. Section 2 describes its statistical model (a nonparametric model based on first order moments) and the clustering algorithm derived from it (Algorithm 1) is presented. Algorithm 1 is extremely simple (needless to say, this simplicity is a virtue of the proposed method), and is only a slight modification of many of the standard agglomerative clustering methods. However, despite its simplicity, the deep insights of the algorithm are explained in Section 3. Remarkably, very roughly speaking, the affinity of each hierarchical cluster is expressed as the height of its dendrogram, which, under certain assumptions, asymptotically approaches its true value. Finally, Section 4 describes the effectiveness of the proposed algorithm on real data, as well as its limitations, and provides code for follow-up studies.

Note: I was not able to understand the details (techniques in the supplementary material) of some of the theoretical analysis in this paper within the initial review period. For Theorem 1, I was able to follow the outline of its key mathematical induction. For Theorem 2, I have not been able to grasp the details. However, the statements in the text are very reliable, and the authors have made an effort to add intuitive explanations to these theoretical results.

**Strengths:**

- This paper is an excellent combination of a simple methodology and deep insights behind it. Needless to say, methodological simplicity is a virtue of the proposed method at this time.
- The phenomenon of affinities between clusters at each level of hierarchy appearing visualized as heights on the dendrogram will be the attractive results for a wide range of readers, both theoretically and in terms of application.
- It is excellent that the theoretical results (Theorem 1 and Theorem 2) are carefully interpreted in Section 3.3 as detailed intuitions.
- Authors also share the code (Jupyter Notebook) for re-experimentation with the community, which is a great contribution to subsequent researchers.
- In Section 5, the paper explicitly mentions the slightly stronger assumption of the proposed model and clarifies for what cases (what data) it has a negative impact. This is an essential finding in the development of science and technology.

**Weaknesses:**

I have not been able to find any convincing weaknesses for this paper in the initial peer review. However, I have some concern about whether the statistical model assumed by the proposed method is causing model speciﬁcation for data with so-called multiple clusters. I have inquired about this concern with the author in the following question. If the concern is resolved, I have not found any major weakness in this paper.

**Questions:**

I am very grateful to authors for sharing very insightful and interesting ideas. I have some questions because this paper contains simple methodology but very surprising (and perhaps intuitively non-trivial) results. These questions are the reason I am not giving this paper a higher score at this time for its overall rating.
These may simply be my lack of insight, but perhaps a brief mention in the text or in the supplementary material would make the advantages of the proposed method clearer to some readers.
I would be very grateful if a response from the authors could clear up some of my questions.

(1) Another aspect of the model assumptions (Section 2.1).

It is very interesting to note that the only model mentioned in this paper is the chain rule by Equation 3, and everything else can be expressed nonparametrically. However, does Equation 3 imply that "all data are i.i.d. samples from a single cluster (a single nonparametric distribution that assumes only first order moments)"? In other words, is there model misspecification occurring for data sets with multiple clusters (e.g., a 2-cluster Gaussian mixture model)? If we use the chaining rule in Equation 3 to marginalize the random variables at each node in turn from the terminal node to the root node of the tree structure, this may be a model in which all data are sampled i.i.d. from some nonparametric distribution with identical mean values (only the first order moments are specified). Is this intuition wrong? If this intuition is correct, then surely the "PCA does not do anything wrong" in the second half of Theorem 2 (Equation 9) makes a lot of sense. Conversely, if the proposed statistical model is such that it has multiple clusters (like a Gaussian mixture model, where each cluster tends to gather data at its center), then it is highly counterintuitive that "still PCA does not do anything bad". In summary, my question comes down to whether the proposed model assumes a one-cluster i.i.d. model.

(2) The validity of greedily searching for the closest affinity pair in row 3 of Algorithm 1.

From the viewpoint of an agglomerative algorithm, it seems quite reasonable to greedily look for the closest affinity pair in line 3 of the algorithm. On the other hand, from the perspective of a statistical model, a probabilistic selection (i.e., Markov chain selection), such as finding a single pair from a categorical distribution weighted by affinity proportions, also seems natural. Does the fact that this row 3 is a greedy selection play an important role in the theoretical analysis in section 3? The actual intent of this question can be attributed to something that is also closely related to question (1), as follows.
For example, if the true data distribution is like a Gaussian mixture model with multiple clusters, is it possible to recover the true data distribution (mixture model structure) by, for example, truncating the tree at some hierarchy (resolution) of the tree structure obtained with the proposed model? In other words, would there not be a misspecification problem as a statistical model? My rough intuition tells me that it is not easy to recover the structure of a statistical model in the case of a greedy merge.
I also know that such a topic is a bit off the author's actual intent for this paper. However, the question arises because I am very curious as to why the statistical model behind the proposed method leads to very good properties (Theorem 1 and Theorem 2) in Algorithm 1.

(3) (Very minor.) Reason why DAG is first introduced.

Is there a reason why when the tree model is introduced in lines 68-69, it is broadly defined as directed acyclic graphs (DAGs) rather than restricted to binary trees?
DAGs are often used as the structure of Bayesian networks. Are there any indications that the theoretical results of this study for binary trees may be extended to DAGs or Bayesian networks?

Finally, once again, I appreciate your sharing of some very interesting ideas.

**Limitations:**

The paper carefully questions the weaknesses of the proposed method in Section 5. It is well explained what kind of data and in what cases the proposed method is unlikely to show its true value.

On another point, I am slightly concerned whether the statistical model in the proposed method implicitly assumes i.i.d. data with a single cluster, as I have asked the authors in my question. However, this concern may be addressed by a response from the authors.

---

> ### Author Rebuttal · Authors · 2023-08-09
>
> > **Weaknesses**
> >> I have not been able to find any convincing weaknesses for this paper in the initial peer review. However, I have some concern about whether the statistical model assumed by the proposed method is causing model speciﬁcation for data with so-called multiple clusters. I have inquired about this concern with the author in the following question. If the concern is resolved, I have not found any major weakness in this paper.
>
> Please see below.
>
> > **Questions:**
> >> - Another aspect of the model assumptions (Section 2.1). ... It is very interesting to note that the only model mentioned in this paper is the chain rule by Equation 3, and everything else can be expressed nonparametrically. However, does Equation 3 imply that "all data are i.i.d. samples from a single cluster (a single nonparametric distribution that assumes only first order moments)"? In other words, is there model misspecification occurring for data sets with multiple clusters (e.g., a 2-cluster Gaussian mixture model)?
>
> Thanks for this thought-provoking question.
>
> We can cast a 2-cluster Gaussian mixture model as a special case of our model, but subject to the Gaussian cluster centers being random and themselves having the same mean. To set this up we need a tree with three vertices $\mathcal{V}=\\{0,1,2\\}$ with $0$ being the root, edges $(0,1)$ and $(0,2)$, and $\mathcal{Z}=\\{1,2\\}$. For simplicity, let $\mathbf{X}(0)$ be a constant (that is, not random), and let $\mathbf{X}(1)$ and $\mathbf{X}(2)$ be iid random vectors, each with mean $\mathbf{X}(0)$. If the vectors $\mathbf{E}_i$, for $=1,\ldots,n$ are Gaussian, then overall $Y_1,\ldots,Y_n$ are draws from a 2-component, Gaussian mixture with random centres, as claimed.
>
> We need $\mathbf{X}(1)$ and $\mathbf{X}(2)$ to have mean $\mathbf{X}(0)$ in order for eq (2) to hold, which is in turn needed to establish the relationship between affinity $\alpha$ and merge height $m$ in Lemma 1. This is really fundamental to our approach.
>
> So, mathematically speaking, if data were generated from a model with deterministic cluster centers, then technically our model of the latent cluster centers would be misspecified. However, if we are only given data $Y_1,\ldots,Y_n$ we believe it is not possible to identify whether cluster centers are random or not, i.e. this technical model misspecification for latent cluster centers does not imply misspecification of the induced model of observed data. Hence, overall we think there is no problem here.
>
> >> - (2) The validity of greedily searching for the closest affinity pair in row 3 of Algorithm 1. From the viewpoint of an agglomerative algorithm, it seems quite reasonable to greedily look for the closest affinity pair in line 3 of the algorithm. On the other hand, from the perspective of a statistical model, a probabilistic selection (i.e., Markov chain selection), such as finding a single pair from a categorical distribution weighted by affinity proportions, also seems natural. ... The actual intent of this question can be attributed to something that is also closely related to question (1), as follows. For example, if the true data distribution is like a Gaussian mixture model with multiple clusters, is it possible to recover the true data distribution (mixture model structure) by, for example, truncating the tree at some hierarchy (resolution) of the tree structure obtained with the proposed model? In other words, would there not be a misspecification problem as a statistical model?
>
> The topic of recovering the data distribution is a fascinating one, but not one we have considered before. We note in passing, it is true that if one knows all the pairwise merge heights of a dendrogram (in out setup) then one can reconstruct the entire tree - this is discussed in appendix 3.C. What you suggest about an alternative to the greedy merge sounds somewhat like the integration performed in the EM algorithm for mixture models, or Bayesian approaches to hierarchical clustering, where the associations between data points and clusters are integrated out. We have not attempted to analyse such approaches in our framework, but this could be an interesting  and new avenue of future research -- perhaps some kind of consistency result for likelihood-based or Bayesian model fitting -- but this is well beyond the scope of the present paper.
>
> >> - (3) (Very minor.) Reason why DAG is first introduced. Is there a reason why when the tree model is introduced in lines 68-69, it is broadly defined as directed acyclic graphs (DAGs) rather than restricted to binary trees? DAGs are often used as the structure of Bayesian networks. Are there any indications that the theoretical results of this study for binary trees may be extended to DAGs or Bayesian networks?
>
> The point of not restricting the binary trees is just to emphasise that our theory tells us the algorithm can produce useful output, without assuming such binary structure, even though the algorithm itself involves binary merges. We haven't yet considered more general Bayesian networks, but again this could be an interesting and new avenue for future research.
>
> Thanks very much for these stimulating questions!

---

> > ### Comment · Reviewer_Emfa · 2023-08-17
> > **Thank you for your detailed and insightful answers.**
> >
> > Thank you for your detailed and insightful answers. The author has given satisfactory answers to my three questions. I now have a better understanding of the current value of this study and new perspectives on future research. I would like to maintain my score as per my initial positive impression.

---

### Official Review · Reviewer_wZ8j · 2023-07-05

**Soundness:** 3 good
**Presentation:** 2 fair
**Contribution:** 3 good
**Rating:** 5
**Confidence:** 3

**Summary:**

This paper presents new analysis and perspectives on one of the most widely used clustering algorithms, hierarchical agglomerative clustering (HAC). In particular, the authors consider the relationship between a particular generative process of data and a dot-product based linkage of HAC. Empirical and theoretical results are presented.

**Strengths:**

This paper presents an interesting perspective on a widely studied clustering algorithm.

I don't think that the intention of the paper is to yield surprise that for certain kinds of data there might be one linkage function (dot product) that works better than others. Rather I think the intention of the paper is to draw connections between clustering models and HAC.

In particular, I see the strengths as:

* Connections between the model for latent tree structures in Eq. 1 and Theorem 2 and dot-product based average linkage
* Connections between models for trees with heights and HAC.
* Empirical analysis of some of the theoretical scaling properties: Figure 2.


**Weaknesses:**

I think that the paper could be improved in the following ways:

* W1. I think that more explicit treatment of Eq. (2) and (3) would improve the presentation; e.g., showing where/why these hold under the model in Eq. (1).
* W2. My understanding is that Lemma 4 & Prop 1 looks quite similar to standard HAC proofs about reducibility? I think it would greatly improvement the treatment of the result to explain the differences / distinctions. Apologies if I have missed something.
* W3. While the evaluation metric seems quite reasonable, might also be interesting to show results across other metrics such as dendrogram purity or similar metrics. I see this as a minor point though.
* W4. In the advent of distributed (typically relatively low) dimensional representations from deep neural encoder models. I wonder if the authors could provide perspective on "theorem 2 says that affinity estimation error vanishes if the dimension p grows faster enough relative to n". E.g., should we think of this method as appropriate in such circumstances where p is likely in range of 128-1024 or so and n might be in the millions or billions?

**Questions:**

* Q1. How could the presentation be modified to address W1?
* Q2. (W2) How does reducibility related to the proofs of Lemma 4 / Prop 1?
* Q3. (W3) did you consider using any other metrics for evaluation? If you were to add another metric to provide a new slice of information, which metrics/measurements would you want to add?
* Q4. Please see W4



**Limitations:**

Yes.

---

> ### Author Rebuttal · Authors · 2023-08-09
>
> > **Weaknesses**
> >> - W1. I think that more explicit treatment of Eq. (2) and (3) would improve the presentation; e.g., showing where/why these hold under the model in Eq. (1).
>
> (2) and (3) are not a consequence of (1), but rather distributional assumptions we make about the ingredients of (1), we would happily clarify this in the manuscript. We could make (2) and (3) numbered assumptions, of the form (Ax), if that would be clearer.
>
>
> >> - W2. My understanding is that Lemma 4 & Prop 1 looks quite similar to standard HAC proofs about reducibility? I think it would greatly improvement the treatment of the result to explain the differences / distinctions. Apologies if I have missed something.
>
> The arguments involved in the proofs of Lemma 4 & Prop 1 are indeed similar in flavour to those concerning reducible linkage functions in e.g.:
>
> Sumengen, Baris, et al. "Scaling hierarchical agglomerative clustering to billion-sized datasets." arXiv preprint arXiv:2105.11653 (2021).
>
> and historical references therein. However we are not aware of any results in the literature which give us precisely what we need for Lemma 4 and Prop 1, and indeed we derived them from scratch. We could of course clarify this in the manuscript with appropriate references.
>
> >> - W3. While the evaluation metric seems quite reasonable, might also be interesting to show results across other metrics such as dendrogram purity or similar metrics. I see this as a minor point though.
>
> Thanks for this suggestion. Looking up dendrogram purity in:
>
> Heller, Katherine A., and Zoubin Ghahramani. "Bayesian hierarchical clustering." Proceedings of the 22nd international conference on Machine learning. 2005.
>
> it appears to be applicable in cases where the ground-truth labels specify a partition of the data points, i.e. a "flat" clustering, but are not hierachical in nature. By contrast, the tau-B correlation measure in our manuscript is constructred to compare estimated with ground-truth hierarchical labelling, and indeed our theoretical results concern the quality of estimated hierarchy.  We agree it could be interesting to consider other metrics, but the connection of such metrics to our theory might be quite unclear if they don't directly quantify hierarchy recovery. This could be an avenue for future investigation.
>
> >> - W4. In the advent of distributed (typically relatively low) dimensional representations from deep neural encoder models. I wonder if the authors could provide perspective on "theorem 2 says that affinity estimation error vanishes if the dimension p grows faster enough relative to n". E.g., should we think of this method as appropriate in such circumstances where p is likely in range of 128-1024 or so and n might be in the millions or billions?
>
> The polynomial moment parameter $q$, appearing and the convergence rate $n^{2/q} / p^{1/2}$ in theorem 2 is key here. Under assumption A2, $q$ quantifies how light/heavy the tails of the distributions of the data are -- higher $q$ corresponding to lighter tails. In the numerator of $n^{2/q} / p^{1/2}$ we see increasing $q$ acts to ameliorate the effect of increasing $n$. Futhermore, as mentioned on line 272 in section 4.2, in line with our empirical findings (fig. 2) we conjecture that when A2 is strengthened from polynomial to exponential-of-quadratic (i.e. sub-Gaussian) moments, the convergence rate improves from $n^{2/q} / p^{1/2}$ to $(\log_e  n)^{1/2}  / p^{1/2}$. This means that, theoretically, convergence would occur as long as $p$ grows faster than $\log_e n$, in which case $p$ in the range $128-1024$ you mention could be enough to counter-act $n$ in the range $10^6 - 10^9$.
>
> >**Questions:**
> >> - Q1. How could the presentation be modified to address W1?
>
> See above.
>
> >> - Q2. (W2) How does reducibility related to the proofs of Lemma 4 / Prop 1?
>
> See above.
>
> >> - Q3. (W3) did you consider using any other metrics for evaluation? If you were to add another metric to provide a new slice of information, which metrics/measurements would you want to add?
>
> Further to our response on this topic above, to illustrate the appliability of our theory in greater detail, it would be desirable to have ground-truth merge heights available, so that estimated versus ground-truth merge heights could be used as a metric. Such ground-truth merge heights might potentially be available in some phylogenetic application domains, and in future we hope to engage with experts in such domains to investigate this further.  If in future we could extend out theoretical results from merge-height estimation to quantifying the quality of "flat" clusterings derived from those merge heights, then the dendrogram purity measure you suggest could be an interesting addition.
>
> >> - Q4. Please see W4
>
> See above.

---

> > ### Comment · Reviewer_wZ8j · 2023-08-20
> >
> > Thanks very much for your response. I appreciate the additional clarifications.
> >
> > * I indeed think that writing (2) and (3) as numbered assumptions would be better. Along with examples where assumptions hold and do not hold.
> > * I think that mentioning reducibility in the discussion of the proofs would be an improvement to the paper.
> > * I agree with your assessment of dendrogram purity.
> > * Your comment for W4 is quite helpful, thank you. I think it would improve the paper to add this and perhaps more (e.g., more emphasis on this on Figure 2) be added as a remark (at least to supplement).

---

### Official Review · Reviewer_FShC · 2023-07-06

**Soundness:** 3 good
**Presentation:** 2 fair
**Contribution:** 3 good
**Rating:** 7
**Confidence:** 3

**Summary:**

The authors discuss a phylogenetic reconstruction problem (I'm not 100% clear on the exact problem, though) and suggest to use the dot product as a measure of similarity (or affinity). In particular, they seem to use the UPGMA algorithm (Alg. 1) where similarity is defined by dot product (scaled by $1/p$).

The proposed method is shown to recover the underlying tree structure with guaranteed accuracy under reasonable theoretical assumptions. Empirical results indicate good performance in a number of cases where the ground-truth hierarchy is known.

**Strengths:**

- the theoretical guarantees appear very strong

**Weaknesses:**

- presentation should be improved so that the problem under study is clearer
- it seems that methodologically, there is no novelty beyond proposing to use the dot product as a similarity measure in the UPGMA method
- limited set of benchmark methods in the empirical part

**Questions:**

I am familiar with phylogenetics and probabilistic graphical models, but despite reasonable efforts, I fail to be able to comprehend the modeling setup. The authors should clearly explain what the model in Eq. (1) means. In particular, the meaning of the latent variables $Z_i$, and the two mappings, $X(Z_i)$ and $S(Z_i)$, that generate the observed data from the latent variables and the random noise, $E_i$, should be explained.

Another general thing I was left wondering is the logic behind the dot product as an affinity measure. It is common to use the cosine similarity $\cos \theta = \frac{\langle x, y\rangle}{||x|| ||y||}$, where $\theta$ is the angle between the vectors $x$ and $y$, and $\langle x, y\rangle$ is the dot product between them. In other words, the cosine similarity is the dot product scaled by the product of the Euclidean norms of the two vectors. Such scaling can be motivated by noting that without normalization, the dot product has the peculiar property that, for instance, $\langle Cx, x\rangle = C \langle x, x\rangle$, i.e., for $C>1$, the vectors $Cx$ and $x$ are more similar (in terms of the dot product) to each other than the vectors $x$ and $x$ (so the vector $x$ to itself). For normalized vectors, the dot product and the cosine similarity are identical. (Further, for normalized vectors, the Euclidean distance equals $\sqrt{2 - 2{\langle x, y\rangle}}$ so it too is a monotonic function of the dot product.)  I would appreciate some comments on the motivation of using the dot product as opposed to commonly used metrics such as the Euclidean distance or the cosine similarity, and on when it is likely to be appropriate (and when not).

detailed comments:
- I suppose some would prefer to use the $x \cdot y$ notation for dot product since $\langle x,y \rangle$ often denotes the (more general) inner product
- p. 2: "distributional properties $\mathbf{X}$": should this be "... of $\mathbf{X}$"?
- p. 7: When you say you compare against the UPGMA method, you should say what distance metric you use in UPGMA. In fact, it would be interesting to see the results of UPGMA (and other methods) based on various distance metrics (dot product, cosine, Euclidean, ...). And indeed, why not include more phylogenetic methods such as Neighbor-Joining (NJ), etc?
- appendix C, proof of Lemma 2: the second line of the displayed equation should have a factor 2 in front of the dot product term (as in $||x+y||^2 = ||x||^2 + 2 \langle x,y \rangle + ||y||^2$)

**Limitations:**

See above

---

> ### Author Rebuttal · Authors · 2023-08-09
>
> >**Weaknesses**
> > - presentation should be improved so that the problem under study is clearer
>
> See response to questions below.
>
> > - it seems that methodologically, there is no novelty beyond proposing to use the dot product as a similarity measure in the UPGMA method
>
> Indeed beyond the point of using the dot-product affinity measure, we are not claiming methodological novelty. Rather our contribution is to put forward an entirely new perspective on a very well known and very widely used type of algorithm.
>
> > - limited set of benchmark methods in the empirical part
>
> Please see below for clarifications - especially regarding UPGMA and the additional numerical results in appendix D.
>
> >**Questions**
> > - I am familiar with phylogenetics and probabilistic graphical models, but despite reasonable efforts, I fail to be able to comprehend the modeling setup. The authors should clearly explain what the model in Eq. (1) means.
>
> We would happily add some commentary. For example,  for each $v\in\mathcal{Z}$ one can think of the vector $\mathbf{X}(v)\in\mathbb{R}^p$ as the random centre of a "cluster", with correlation structure of the cluster determined by the matrix $\mathbf{S}(v)$. The latent variable $Z_i$ indicates which cluster the ith data vector $\mathbf{Y}_i$ is associated with. The vectors $\mathbf{X}(v)$, $v\in\mathcal{V}\setminus\mathcal{Z}$, correspond to unobserved vertices in the underlying tree.
>
> > -Another general thing I was left wondering is the logic behind the dot product as an affinity measure.  It is common to use the cosine similarity ... I would appreciate some comments on the motivation of using the dot product as opposed to commonly used metrics such as the Euclidean distance or the cosine similarity, and on when it is likely to be appropriate (and when not).
>
> We wonder if the reviewer may have missed appendix E in the supplementary material, "Understanding agglomerative clustering with Euclidean or cosine distances in our framework". We refer to appendix E on lines 46, 139, 232, 256 and 299 of the main part of the manuscript.
>
> Section E.1 discusses use of Euclidean distance as a similarity meausre and section E.2 makes the connection between our algorithm and using cosine distance. Theorem 3 in section E.3 shows that using cosine distance as a similarity measure can work well under an alternative model in which errors are multiplicative, rather than additive as in eq. (1) in the main part of the manuscript, and the "error-free" data vectors $\mathbf{X}(Z_i)$ all have the same expected square magnitude. Contrasting this with theorem 2 highlights that the dot-product affinity is a good choice to help remove additive noise, and can cope with data where expected square magnitudes vary across data points - see also figure 1 of the main manuscript for interpretation of dendrogram height.
>
> In appendix E.4 we discuss the limitations of our modelling assumptions and failings of the dot-product affinities.
>
> In connection with what the reviewer writes about $\langle Cx,x\rangle,$ we had thoughts along the same lines: we chose the word dot-product "affinity" rather than "similarity" purposefully;  indeed it is possible that $\langle Cx,x\rangle >\langle x,x \rangle$, whereas it would be confusing to suggest that $Cx$ and $x$ are more "similar" to each other than $x$ is to itself. We would happily add a note to explain this in the manuscript.
>
> > - I suppose some would prefer to use the $\cdot$ notation for dot product since often denotes the (more general) inner product.
>
> We see your point here and have experimented with other notation, but found the "$\cdot$"" notation to be less visually clear in many equations.
>
> > - p. 2: "distributional properties ": should this be "... of "?
>
> Thanks for catching this.
>
> > - p. 7: When you say you compare against the UPGMA method, you should say what distance metric you use in UPGMA. In fact, it would be interesting to see the results of UPGMA (and other methods) based on various distance metrics (dot product, cosine, Euclidean, ...). And indeed, why not include more phylogenetic methods such as Neighbor-Joining (NJ), etc?
>
> We had used "UPGMA" to refer specifically to the combination of average linkage and Euclidean distance, in retrospect we realise this is confusing. In fact results for average linkage with cosine distance (i.e. UPGMA with cosine distance) are already given in table 1 in the main manuscript (there labelled "Cosine distance"). We could clarify this.
>
> Following the reviewers suggestion, in the .pdf attached to this response we give numerical results for UPGMA with Manhattan distance, and UPGMA with dot-product as a dissimilarity measure (hence "opposite" to our Algorithm 1). Performance of these algorithms is worse than that of Algorithm 1.
>
> We wonder if the reviewer may not have seen the additional numerical results in table 2 in appendix D of the supplementary material, where we compare our algorithm against other combinations of cosine and Euclidean distances, with complete and single linkage functions. In total across the main manuscript and appendix D there are comparisons of our method against 8 others:
>
> main manuscript, table 1:
> - "Cosine distance" (i.e. UPGMA with cosine distance)
> - HDBSCAN
> - "UPGMA" (i.e. UPGMA with Euclidean distance)
> - Ward's method
>
> appendix D, table 2:
> - Complete linkage with Euclidean distance
> - Complete linkage with Cosine distance
> - Single linkage with Euclidean distance
> - Single linkage with Cosine distance
>
> so including the new UPGMA results, the total number of algorithms we compare against is 10.
>
> Thanks for the suggestion about neighbour-joining. Our theory and performance measure concern hierarchy recovery, and in the time available we haven't found an implementation of neighbour-joining which allows hierarchy (rather than a "flat" clustering) to be extracted. With more time we could investigate this further.
>
> > - appendix C, proof of Lemma 2: the second line...
> Thanks for catching this.

---

> > ### Comment · Reviewer_FShC · 2023-08-15
> > **Still not following the intuition about dot product affinity -- but overall, changing my rating to accept**
> >
> > Thanks for the response.
> >
> > The clarification about the underlying model is helpful. However, I'm still equally puzzled by the intuition behind usingt the dot product (rather than its normalized version, the cosine similarity) as an affinity measure. I had indeed not read Appendix E since I expect that any content that is essential for understanding the main ideas presented in the paper are included in the main paper (otherwise one can asek what is the point of the page limit). In Appendix E, the authors provide some remarks on the relationship between dot product and cosine similarity -- concluding, e.g., that under norm constraints, cosine similarity and dot product affinity approximate each other (Thn. 3), which is (on an informal level) obvious as pointed out in my review above. The authors interpret this as showing that the cosine similarity can work well under such constraints. Still, it doesn't help in building intuition about why dot product would be a sensible choice in cases where the norms are not nearly uniform.
> >
> > In any case, having also read the other reviews and the rebuttals related to them, I can only conclude that the theoretical results (mainly Thm. 1) seem to support the authors' claims about the good performance, and since this is (at least to my intuition) somewhat unexpected, I believe the paper is worth publishing and exposing to the wider community's evaluation.
> >
> > Considering all of the above, I am changing my rating to accept.

---

> > > ### Comment · Reviewer_FShC · 2023-08-21
> > > **PS. on NJ**
> > >
> > > lastly, wanted to add that I find it hard to believe that the authors couldn't find a Neighbor-Joining implementation that outputs hierarchical clustering instead of flat ones: all implementations that I can find give hierarchical clusterings (or trees). E.g., based on a five minute googling:
> > > * http://scikit-bio.org/docs/0.2.1/generated/skbio.tree.nj.html
> > > * https://rdrr.io/cran/ape/man/nj.html
> > > * https://www.biotite-python.org/apidoc/biotite.sequence.phylo.neighbor_joining.html
> > > * https://pypi.org/project/TreeMethods/
> > > * https://biopython.org/wiki/Phylo

---

> > > > ### Author Response · Authors · 2023-08-21
> > > > **Concerning neighbor joining and unrooted trees.**
> > > >
> > > > Thanks very much for these links, we did see these methods.
> > > >
> > > > To clarify, the issue is that neighbor joining, by default, outputs an unrooted tree, see e.g., the first link you gave:
> > > >
> > > > http://scikit-bio.org/docs/0.2.1/generated/skbio.tree.nj.html
> > > >
> > > > Our model, algorithm, theory and performance measure all concern rooted trees and rely heavily on the notion of most recent common ancestor.
> > > >
> > > > For unrooted trees, most recent common ancestor is not defined. If one converts an unrooted tree to a rooted tree by choosing a root, most recent common ancestors then depend on that choice of root.
> > > >
> > > > Methods such as those you point to include various different techniques for choosing a root, but in the time available it’s not been clear to us how to approach root choice in a principled and transparent way for purposes of hierarchy recovery and comparison.

---

> > > > > ### Comment · Reviewer_FShC · 2023-08-21
> > > > > **On NJ and ranking**
> > > > >
> > > > > I understand. But my next question is, isn't your performance metric (Kendall ranking correlation) applicable to unrooted trees just as well as rooted trees? I mean, what you need to get from the tree is, for each node, a ranking "according to the order in which [the other nodes] merge with [the node of interest]". You can get this from an unrooted tree by calculating the distance between the node of interest and each of the other nodes. NJ returns trees with edge lengths, so you can define the distance based on them.

---

> > > > > > ### Author Response · Authors · 2023-08-21
> > > > > >
> > > > > > The short answer to the reviewer's question is:
> > > > > >
> > > > > > "No, the performance metric is not applicable to unrooted trees."
> > > > > >
> > > > > > The order in which nodes merge with the node of interest would change if we were to change the root of the tree. The method of ranking nodes by graph distance from the node of interest which the reviewer suggests is not equivalent.

---

### Official Review · Reviewer_s6Xk · 2023-07-10

**Soundness:** 3 good
**Presentation:** 3 good
**Contribution:** 3 good
**Rating:** 6
**Confidence:** 4

**Summary:**

The paper studies hierarchical agglomerative clustering under a specified generative process for the underlying data vectors. The main focus of the paper is similarity based clustering that deals with inner products between vectors rather than computing pairwise distances that has been studied before.

The goal of the paper is to provide recovery guarantees for the underlying tree structure. The paper provides such guarantees under their specified generative model and they bound the maximum merge distortion  by the affinity estimation error (see Theorem 1). Moreover, they provide tradeoffs for the estimation error based on the dimension of the data vectors and the sample size. Finally, the authors test their method in terms of runtime and quality against several other methods from the literature.



**Strengths:**

+hierarchical tree recovery is not a well-understood problem so the overall research direction is interesting
+dot products are used a lot in real-world applications so having analyses based on them is important
+the flavor of the guarantees seem to be towards the right direction for what we would mean "hierarchical tree recovery"
+the technical aspects of the proofs

**Weaknesses:**

-The generative model should be better explained and compared to other previously studied models for hierarchical clustering. In particular, it was not clear to me how to think of your model given that Hierarchical Stochastic Block Models and well-clusterable graphs are defined and seem more natural to me, see e.g. papers below:

Cohen-Addad, Kanade, Mallmann-Trenn, Mathieu: "Hierarchical clustering: Objective functions and algorithms"
Chatziafratis, Mahdian, Ahmadian: "Maximizing Agreements for Ranking, Clustering and Hierarchical Clustering via MAX-CUT"
Manghiuc, Sun: "HierarchicalClustering: O(1)-Approximation for Well-Clustered Graphs"

So overall, having a better exposition and examples for why this particular model you proposed is a natural one, would help the readers a lot.

-The difference from other papers that focus on distance-based hierarchical clustering is important for the mathematics, but conceptually the algorithm is very related. So here the innovation in terms of conceptual contribution is slightly weakened. However, from technical viewpoint there are interesting ideas the paper introduces.

-In the statement of the Theorems 1 and 2, I missed some interpretaion of the results and especially some comparison to related work on distance-based methods. Do we learn something interesting by this dot-product analysis? How are the dimension/sample/estimation tradeoffs different that those other papers?

-The numerical experiments: I am not sure they convey the benefits of your algorithm in terms of estimation error and recovery. In Table 1, I would clarify what are the most significant advantages of your algorithm as currently it is hard to grasp.




**Questions:**

From above:
Q1: Do we learn something interesting by this dot-product analysis and how are the dimension/sample/estimation tradeoffs different that those other papers that did distance-based analyses?
Q2: Perhaps easy, but giving more evidence to motivate your particular hierarchical tree model would be appreciated.
Q3: Technical question: Perhaps to help illustrate some of your techniques/ideas, it would be helpful to instantiate your model to a small-depth hierarchy. Say there are only 2 or 3 levels like you had in some of your experimental datasets (S&P500, 20Newsgroups etc). Can your theorems be more interpretable in this scenario?

**Limitations:**

I like the paper and I find that the authors addressing the concerns mentioned above will significantly improve their paper.
I don't find any major limitations, apart from better motivating their model, comparing it with the existing ones, and better illustrating their Theorems and what we learn from them (in comparison to distance-based methods).

---

> ### Author Rebuttal · Authors · 2023-08-09
>
> > The generative model should be better explained and compared to other previously studied models for hierarchical clustering. ... see e.g. papers below ... having a better exposition and examples for why this particular model you proposed is a natural one, would help the readers a lot.
>
> Thanks for these references. In fact we already cite Cohen-Addad et al. "Hierarchical clustering: Objective functions and algorithms" in the last paragraph of section 1, it is item 14 in our bibliography. The paper of Manghiuc et al. falls within the cost-function based framework of Dasgupta (bibliograpy item 15) cited and discussed in section 1. There we note that this existing modelling approach involves assuming an underlying ultrametric space whose geometry specifies the unknown tree. The paper of  Chatziafratis et al. you point to is closely related to the paper of Charikar and Chatziafratis "Approximate Hierarchical Clustering via Sparsest Cut and Spreading Metrics" (bibliography item 11) also cited in section 1. There we highlight that high-dimensionality $p\to\infty$ is a key consideration in our analysis, whereas in existing works dimension is either fixed, or not considered at all.
>
> Regarding the matter of how to think of our model, our entire setup is really quite different and new compared to existing works, so direct comparisons may not be very meaningful. However, one key feature is that the "ground truth" tree structure underlying our model is the conditional independence graph (see eq. (2) in the manuscript), as opposed to e.g. the geoemtry of an ultrametric as mentioned above. This is really a new perspective, and we think it is important because of the fundamental role that conditional independence plays in hierarchical statistical modelling.
>
> Regarding exposition, we intended fig. 1 to help the reader understand how this tree structure relates to data, via the specific notion of dendrogram which we introduce in the paper. Regarding examples, one way we could add some content here is to expand upon the simulation example described in appendix D, which we use in our numerical results in section 4.
>
> We would like to clarify a somewhat subtle point about our objectives: it is _not_ our main objective to specify a model which we necessarily believe is more natural than those in other works. Rather, our intention is to uncover the fact that a fairly standard algorithm, albeit using a dot-product affinity, allows the conditional independence tree underlying data to be recovered, under assumptions which we believe are quite general.
>
> > - The difference from other papers that focus on distance-based hierarchical clustering is important for the mathematics, but conceptually the algorithm is very related. So here the innovation in terms of conceptual contribution is slightly weakened.
>
> Indeed we are not claiming significant methodological innovation, but rather a new perspective on a well-known type of algorithm.
>
> >  - In the statement of the Theorems 1 and 2, I missed some interpretaion ... especially some comparison to related work on distance-based methods. Do we learn something interesting by this dot-product analysis? How are the dimension/sample/estimation tradeoffs different that those other papers?
>
> As noted above and in section 1 of the manuscript, in existing works, dimension $p$ is either fixed or does not appear at all, and various forms of convergence are driven by increasing sample size $n\to\infty$, whereas in our setup $p\to\infty$ drives convergence - see eq. (8) and (9). To our knowledge, this is the first time it has been established that high-dimensionality might have a beneficial effect on performance. The convergence rate $n^{2/q}/\sqrt{p}$ in theorem 2 is unusual. At first glance it may appear strange that sample size $n$ appears in the numerator, but this is a reflection of the very conservative $\max$-form of error we are considering on the r.h.s. of (8). Moreover, we see from $n^{2/q}$ that the polynomial moment parameter $q$  from assumption A2 ameliorates the effect of $n$ being large. These are all new insights.
>
> Regarding comparison to distance-based methods, we note that supplementary material appendix E "Understanding agglomerative clustering with Euclidean or cosine distances in our framework" is dedicated to helping make such connections. In E.3 we establish a convergence result (theorem 3) similar to theorem 2 but concerning cosine distance, under a modified model in which noise is multiplicative rather than additive as in eq. (1).
>
> > - The numerical experiments: I am not sure they convey the benefits of your algorithm... . In Table 1, I would clarify what are the most significant advantages of your algorithm as currently it is hard to grasp.
>
> Thanks for this suggestion, the basic message here is that our algorithm does a better job of recovering ground-truth hierarchy, except for with the S/&P500 data - shortcomings of our modelling approach which may explain why are discussed in appendix E.4.
>
> >**Questions**
> > - Q1: Do we learn something interesting by this dot-product analysis and how are the dimension/sample/estimation tradeoffs different that those other papers that did distance-based analyses?
>
> See above.
>
> > - Q2: Perhaps easy, but giving more evidence to motivate your particular hierarchical tree model would be appreciated.
>
> Also see above.
>
> > - Q3: Technical question: Perhaps to help illustrate some of your techniques/ideas, it would be helpful to instantiate your model to a small-depth hierarchy. ... Can your theorems be more interpretable in this scenario?
>
> Thanks for this thought-provoking question. Having considered this, we think that the number of levels as you suggest does enter very directly in to our theoretical framework, and so we can't see an obvious simplification here. This may seem suprising, but we believe it is related to the fact we quantfy performance with merge heights, pair-wise between data points.

---

> > ### Comment · Reviewer_s6Xk · 2023-08-19
> > **Author Response**
> >
> > I would like to thank the authors for their detailed response. My initial positive score remains the same.

---

### Official Review · Reviewer_dt3f · 2023-07-19

**Soundness:** 4 excellent
**Presentation:** 4 excellent
**Contribution:** 3 good
**Rating:** 8
**Confidence:** 3

**Summary:**

The paper discusses a new perspective on hierarchical clustering using the dot product as similarity measure instead of some distance. Under mild conditions on the probabilistic graphical model that generates the data, the proposed algorithm is shown to faithfully recover the underlying tree geometry. Surprisingly, the theoretical results show that the performance of the approach does not suffer, but instead benefits from high-dimensional data.

**Strengths:**

The paper is well written and all theoretical results and definitions are accompanied by intuitive explanations (e.g., lines 84-90, lines 159-166, Fig. 1, or Sec. 3.3). This makes the paper accessible to an audience not versed in graphical models, for example. The results seem quite promising; it is noteworthy that the authors also included "negative" results for the S&P500 dataset, and that they made an effort in investigating the limitations of the proposed method.

**Weaknesses:**

I see very little weaknesses in the paper (see below for a list of typos). There are only few points unclear (see Questions below). The only concern I have is that the results claim that the performance of the method improves if the dimensionality of the data increases. This is probably caused by the mixing condition, which ensures that more dimensions deliver more relevant information. This assumption is quite questionable in my opinion, as usually more dimensions do not reveal substantially more information about the underlying distribution. As an example, consider natural images: going from low resolutions to higher resolutions may at first improve clustering performance, but at some point increasing the resolution will bring only diminishing returns. The reason is that, as the resolution increases, also the correlation between dimensions increases, which is in contrast to A1. I would appreciate seeing results based on, e.g., lower numbers of TF-IDF features to understand how this affects clustering performance.

Another concern is that Fig. 3a and 3c seem to be inconsistent. In 3a, at the position of comp.windows.x, there are three dark blue squares, one light blue square, and (maybe?) a hidden pink x. In 3c, the top five topics contain two dark blue and two light blue squares. Please check if there was a mistake in generating the figure.

## Minor:
- line 56: "interpretation to in all..."
- line 78: "distributional properties OF $\mathbf{X}$"


**Questions:**

- In line 102, it is not clear why $\mathbb{E}[< Y_i, Y_j>|Z_1,\dots,Z_n]$ is conditioned on all $Z_n$
- What is the meaning of the operator $O_{\mathbb{P}}$?
- In Table 1, the results for simulated data do not depend on whether PCA was applied or not. Is this caused by the fact that the dimensions are independent?
- Since the PCA version of the algorithm operates on fixed dimension $r$, it is not clear to my why in (9) an increase of $p$ should bring any benefit. Could you expand on that?

**Limitations:**

The authors clearly outline the limitations of their study, which is highly appreciated. A possible further limitation of the method is that, apparently, good performance guarantees can only be given for high-dimensional data, which is slightly counterintuitive. Societal implications are not to be expected.

---

> ### Author Rebuttal · Authors · 2023-08-09
>
> >The only concern I have is that the results claim that the performance of the method improves if the dimensionality of the data increases. This is probably caused by the mixing condition ... I would appreciate seeing results based on, e.g., lower numbers of TF-IDF features to understand how this affects clustering performance.
>
> Thanks for this insightful comment. As requested, in the .pdf attached to this response, table 1 and figure 1, we give additional numerical results showing the effect of the number of TF-IDF features, i.e. varying $p$. These results were obtained by randomly choosing features to exclude, to give the desired $p$. We find that, as $p$ grows performance improves, without evidence of "diminshing returns". We note a slight increase in the standard error with $p$, but the standard errors are roughly 100 times smaller than the values of the performance measure (note the factor of $10^3$), so we think this is not a serious concern.
>
> The mixing condition is indeed a key assumption. However, it is not the only reason that performance improves with dimension. To see why, note that from eq. (1),
>
> $$
> p^{-1}\langle\mathbf{Y}_i,\mathbf{Y}_j\rangle = p^{-1}\langle\mathbf{X}(Z_i),\mathbf{X}(Z_j)\rangle + p^{-1}\langle\mathbf{X}(Z_i),\mathbf{S}(Z_j)\mathbf{E}_j\rangle + p^{-1}\langle\mathbf{X}(Z_j),\mathbf{S}(Z_i)\mathbf{E}_i\rangle + p^{-1} \langle \mathbf{S}(Z_i)\mathbf{E}_i,\mathbf{S}(Z_j)\mathbf{E}_j\rangle.
> $$
>
> The mixing assumption allows us to prove $p^{-1}\langle\mathbf{X}(Z_i),\mathbf{X}(Z_j)\rangle-\alpha(Z_i,Z_j)\to 0$ (convergence in probability) as $p\to\infty$, whilst the properties that the "noise" vectors $\mathbf{E}_i$ are zero-mean, have independent entries, are independent of each other, $\mathbf{X}$ and $Z_1,\ldots,Z_n$, allow us to deduce that the second to fourth terms on the r.h.s. of the displayed equation above converge to zero as $p\to\infty$.  In this latter sense, high-dimensionality can help remove noise, irrespective of whether the mixing condition A1 holds.
>
> We agree that for image data as you describe, where $p\to\infty$ corresponds to increasingly high resolution, the mixing assumption is not realistic. Future work would be needed to investigate alternative assumptions in complete detail, but for example, in such situations it could possibly be realistic that $p^{-1}\langle\mathbf{X}(Z_i),\mathbf{X}(Z_j)\rangle-\alpha(Z_i,Z_j)$ is "close" to zero with some high probability for all $p$ large enough, without actually converging to zero. Another interesting possibility is that if one does _not_ assume the mixing condition A1, then one might consider $p^{-1}\langle\mathbf{X}(Z_i),\mathbf{X}(Z_j)\rangle$ as an affinity measure rather than $\alpha(Z_i,Z_j)$. Of course these are just hypotheses, but we believe the work in the manuscript could be a first step towards understanding such scenarios.
>
> For some types of data, such as time series on increasingly long time scales, or geospatial data collected over increasingly large areas, $p\to\infty$ does _not_ correspond higher resolution, and the mixing assumption is arguably realistic. For other data types, such as the document and biological data sets in our paper, it is perhaps less easy to firmly rule the mixing assumption in or out, but we believe our numerical results are encouraging.
>
> >Another concern is that Fig. 3a and 3c seem to be inconsistent. ... Please check if there was a mistake in generating the figure.
>
> The explanation here is that in 3c we do not plot the average dot product affinity between comp.windows.x and itself. However, this is plotted in 3a; the highest dark blue square in the comp.windows.x column corresponds to dot-product with itself. We could update this, or add a comment to clarify.
>
> >**Minor:**...
>
> Thanks for catching these typo's.
>
> >**Questions**
> > - In line 102, it is not clear why  $\mathbb{E}[\langle Y_i,Y_j\rangle|Z_1,\ldots,Z_n]$ is conditioned on all $Z_1,\ldots,Z_n$
>
> Under our model indeed $\mathbb{E}[\langle Y_i,Y_j\rangle|Z_1,\ldots,Z_n]=\mathbb{E}[\langle Y_i,Y_j\rangle|Z_i,Z_j]$, we would happily amend line 102 to the latter.
>
> > - What is the meaning of the operator $O_\mathbb{P}$?
>
> Intuitively, the argument of $O_{\mathbb{P}}(\cdot)$ is the convergence rate. We would happily add a precise defintion of this "big Oh in probability" notation to the manuscript: if $X_{p,n}$ is random variable indexed by $p$ and $n$, then e.g. $X_{p,n}\in O_{\mathbb{P}}\left(n^{2/q} / \sqrt{p}\right)$ means that for any $\epsilon>0$ there exists finite $\delta$ and $M$ such that $n^{2/q} / \sqrt{p}\geq M$ implies:
>
> $$
> \mathbb{P}(|X_{p,n}|>\delta)<\epsilon.
> $$
>
> > - In Table 1, the results for simulated data do not depend on whether PCA was applied or not. Is this caused by the fact that the dimensions are independent?
>
> From a rigorous mathematical point of view, we cannot answer this question definitively: it is possible that a conclusion similar to that in Theorem 2 for PCA holds with the mixing assumption A1 rather than independence assumption, but we haven't proved it. In numerical experiments, we have found that some dependence does not adversely effect PCA, but a detailed examination of this matter would necessitate a longer paper.
>
> > - Since the PCA version of the algorithm operates on fixed dimension, it is not clear to my why in (9) an increase of $p$ should bring any benefit. ...
>
> The proof of theorem 2 and eq. (9) involves analysing the top $r$ eigenvalues and eigenvectors of the $n\times n$ matrix whose $(i,j)$-th entry is $p^{-1}\langle\mathbf{Y}_i,\mathbf{Y}_j\rangle$. As discussed above,  $p^{-1}\langle\mathbf{Y}_i,\mathbf{Y}_j\rangle - \alpha(Z_i,Z_j) \to 0$ when $p\to\infty$. So $p\to\infty$ helps us prove the above mentioned eigenvalues/vectors are close to those of the matrix with elements $\alpha(Z_i,Z_j)$.  For this we need a stronger form of convergence than element-wise matrix convergence, this is where the additional assumption of independence is used.

---

> > ### Comment · Reviewer_dt3f · 2023-08-11
> > **Thanks!**
> >
> > Thank you very much for your detailed answer, especially to my main concern. I agree with your observations that large $p$ helps to decrease noise (even in the absence of mixing), and that mixing is a realistic assumption in many practically relevant cases. I will, for now, keep my score, but may revise it based on the discussion with other reviewers.

---

### Author Rebuttal · Authors · 2023-08-09

The authors are very grateful for the effort which the reviewers have put in to reading our manuscript and providing feedback.

We are pleased to see that, on the whole, the reviewers have recognised and engaged with the new perspective on hierarchical clustering and tree recovery which we report in this paper. Since this perspective is novel in various ways (e.g., in terms of our probabilistic model-based formulation involving conditional independence, emphasis on dot-product affinities, and theoretical investigation of high-dimensional behaviour) it necessitates the use of some concepts and notation which are non-standard in the literature on hierarchical clustering. The authors are very grateful for the various suggestions made by the reviewers which will help us improve the presentation in this regard.

We would like to highlight appendices D and E in the supplementary material of the original submission, in case they were missed. Appendix D.6 reports additional comparative numerical results. Appendix E, entitled "Understanding agglomerative clustering with Euclidean or cosine distances in our framework", provides methodological and theoretical perspectives to help make connections between our framework and agglomerative clustering using more standard distance measures techniques.

Attached to this rebuttal is a .pdf containing additional numerical results in response to the comments from reviewers dt3f and FShC.

---

### Decision · Program_Chairs · 2023-09-21

**Decision:**

Accept (spotlight)

**Comment:**

This paper studies hierarchical clustering via the probabilistic graphical model. The authors propose a simple agglomerative hierarchical clustering algorithm, where clusters are merged by the maximum average dot product, and show its interesting theoretical properties based on the formulation.

This paper is overall well-written, and all the reviewers are happy with the contribution of the paper and recommend acceptance. I also agree with it and like the simplicity of the algorithm with its interesting theoretical analysis.
In particular, this paper gives a new insight into hierarchical clustering using the probabilistic graphical model, which is inspiring and can trigger further development of theoretical studies of hierarchical clustering.

Since the authors made a significant effort on rebuttals, which contain relevant information and successfully address concerns raised by the reviewers, I strongly recommend integrating such contents in the camera-ready version of the paper.

To summarize, I would appreciate the contribution of the paper, which is sound and innovative. Therefore, I recommend the spotlight presentation of the paper.